# SynQ: Accurate Zero-shot Quantization by Synthesis-aware Fine-tuning

**Minjun Kim, Jongjin Kim & U Kang**[*]
Seoul National University, Seoul, South Korea
`{minjun.kim,j2kim99,ukang}@snu.ac.kr`

## Abstract

How can we accurately quantize a pre-trained model without any data? Quantization algorithms are widely used for deploying neural networks on resource-constrained edge devices. Zero-shot Quantization (ZSQ) addresses the crucial and practical scenario where training data are inaccessible for privacy or security reasons. However, three significant challenges hinder the performance of existing ZSQ methods: 1) noise in the synthetic dataset, 2) predictions based on off-target patterns, and the 3) misguidance by erroneous hard labels. In this paper, we propose **SynQ** (**Syn**thesis-aware Fine-tuning for Zero-shot **Q**uantization), a carefully designed ZSQ framework to overcome the limitations of existing methods. SynQ minimizes the noise from the generated samples by exploiting a low-pass filter. Then, SynQ trains the quantized model to improve accuracy by aligning its class activation map with the pre-trained model. Furthermore, SynQ mitigates misguidance from the pre-trained model's error by leveraging only soft labels for difficult samples. Extensive experiments show that SynQ provides the state-of-the-art accuracy, over existing ZSQ methods.

## 1 Introduction

*How can we accurately quantize a pre-trained model without any data?* Despite the success of deep neural networks in various domains, deploying them on resource-constrained edge devices remains challenging due to the limited computing capabilities. Addressing this challenge involves network compression (Cheng et al., 2018; Deng et al., 2020; Park et al., 2024b), where quantization methods (Li et al., 2021; Piao et al., 2022; Gholami et al., 2022) represent the full-precision model with low-bit numbers, achieving high compression rate and accelerated inference with minimal performance degradation compared to other methods such as pruning (Wang et al., 2022; Park et al., 2024a; He & Xiao, 2024), knowledge distillation (Kim et al., 2021; Tran et al., 2022; Cho & Kang, 2022; Jeon et al., 2023a; Xie et al., 2023), and low-rank approximation (Jang et al., 2023). Zero-shot Quantization (ZSQ) (Nagel et al., 2019) further advances this field by permitting quantization without any training data. The importance of this approach is evident in real-world contexts where the training data are unavailable for privacy and security reasons (Sharma et al., 2021).

Among the various existing works (Yoo et al., 2019; Zhang et al., 2021; Guo et al., 2022), methods that fine-tune the quantized model with the synthetic dataset exhibit outstanding performance (Liu et al., 2021a; Zhong et al., 2022b; Fan et al., 2024). Specifically, recent methods generate synthetic samples resembling the real sample distribution by leveraging key aspects from the pre-trained model, such as batch-normalization statistics (Cai et al., 2020), latent embeddings (Choi et al., 2021), or texture feature distribution (Chen et al., 2023). However, we observe that three major limitations still hinder the performance when utilizing synthetic datasets (see Section 3).

- **Noise in the synthetic dataset.** Synthetic datasets have distinct high-frequency noise unlike real images that concentrate on low frequencies (see Figures 1 and 5). This discrepancy results in inefficient fine-tuning of quantized models, thereby directly reducing model performance.

- **Predictions based on off-target patterns.** Quantized model from existing methods rely on incorrect image patterns for predictions (see Figure 2). Such off-target reliance limits the quantized model in identifying key areas necessary for accurate classification.

---

[*]Corresponding Author.

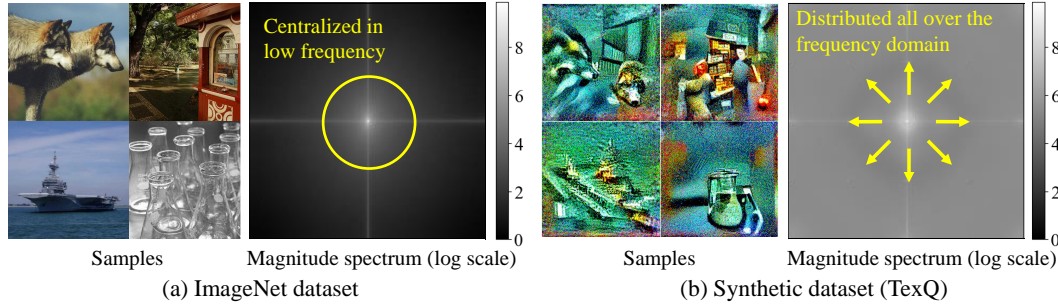

Figure 1: Comparison between (a) real images in ImageNet dataset and (b) generated samples in the synthetic dataset from TexQ (Chen et al., 2023). Each set displays samples labeled as timber wolf, tobacco shop, aircraft carrier, and beaker. We present the average magnitude spectrum for a randomly selected batch of 256 images from each dataset, highlighting their distinct differences.

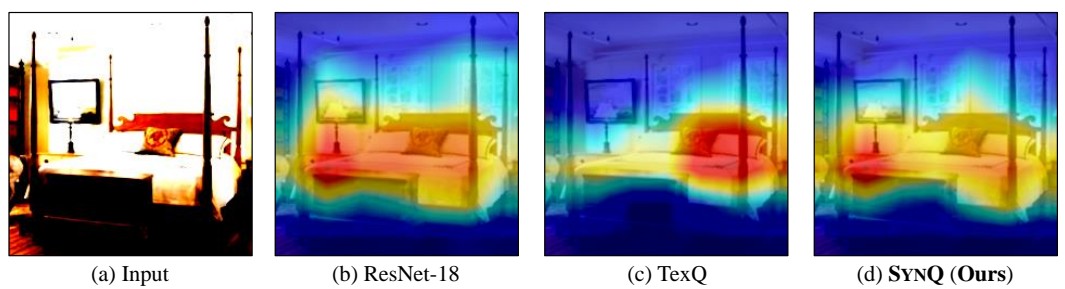

Figure 2: Grad-CAM (Selvaraju et al., 2017) plot of the (a) input by the (b) pre-trained ResNet-18 model on ImageNet dataset, the (c) 3bit quantized model by TexQ, and the (d) 3bit quantized model by SYNQ. While TexQ fails to capture the correct image region, SYNQ captures the region closely matching the pre-trained model.

- **Misguidance by erroneous hard labels.** Hard labels of difficult samples are often incorrect in synthetic dataset, leading to misguided fine-tuning and ultimately harming the model (see Figure 3).

We propose **SYNQ** (**Syn**thesis-aware Fine-tuning for Zero-shot **Q**uantization), an accurate ZSQ fine-tuning method to overcome the limitations of the existing methods that fine-tune with synthetic datasets. SYNQ clears noise from the generated samples within the synthetic dataset by applying a low-pass filter. Then, SYNQ ensures that the quantized model predicts from the correct image region by optimizing the class activation map (CAM) alignment loss to distill object localization knowledge. Furthermore, SYNQ mitigates misguidance from errors of the pre-trained model by using only soft labels for difficult samples. Experimental results show that SYNQ achieves the state-of-the-art performance, improving the image classification accuracy of the quantized model up to 1.74%p compared to existing methods (see Table 1). SYNQ is both powerful and versatile, seamlessly integrating into any ZSQ methods that fine-tune with synthetic datasets, regardless of model type, quantization bits, or dataset (see Sections 5.2, 5.3, Appendices C.6, and C.7).

Our contributions are summarized as follows:

- **Observation.** Our observations clearly outline three significant challenges faced by existing ZSQ methods utilizing synthetic datasets: 1) noise in synthetic datasets, 2) predictions based on off-target patterns, and 3) misguidance by erroneous hard labels (see Figures 1, 2, 3, and 5).
- **Algorithm.** We propose SYNQ, an accurate ZSQ method to overcome the limitations of fine-tuning with synthetic datasets. SYNQ exploits a low-pass filter to minimize noise, aligns the class activation map to ensure prediction from the correct image region, and leverages soft labels for difficult samples to prevent misguidance from erroneous hard labels (see Section 4).
- **Experiments.** We experimentally show that SYNQ consistently outperforms existing ZSQ methods on various models and datasets, achieving classification accuracy improvement of up to 1.74%p (see Section 5 and Appendix C).

**Reproducibility.** All of our implementation and datasets are available at `https://github.com/snudm-starlab/SynQ`.

## 2 PRELIMINARIES AND PROBLEM DEFINITION

We introduce the ZSQ (Zero-shot Quantization) problem and describe the preliminaries. Appendix A contains the detailed descriptions of frequently used notations in this paper.

### 2.1 ZERO-SHOT QUANTIZATION

In this work, we follow the typical two-step scheme (Choi et al., 2021; Li et al., 2023a) to quantize a pretrained model. First, we generate the synthetic dataset that resembles the original dataset using the pre-trained model. Second, we fine-tune the quantized model with generated samples.

The goal of the first step is to produce synthetic dataset $\{\mathbf{x}_i\}_{i=1}^N$ of length $N$ with corresponding labels $\{\mathbf{y}_i\}_{i=1}^N$, using the pre-trained model with parameters $\theta$. We utilize noise optimization (Cai et al., 2020; Zhong et al., 2022b), where we initialize the synthetic dataset and labels as random Gaussian noises and randomly assigned classes, respectively; we then iteratively update the synthetic dataset $\{\mathbf{x}_i\}_{i=1}^N$. Specifically, we minimize Batch Normalization Statistics (BNS) loss $\mathcal{L}_{BNS}$ and Inception Loss (IL) $\mathcal{L}_{IL}$ in Equation (1), with hyperparameter $\alpha$ balancing them.

$$
\min_{\{\mathbf{x}_i\}_{i=1}^N} \mathcal{L}_{IL} + \alpha \mathcal{L}_{BNS}, \quad \text{where} \quad \mathcal{L}_{IL} = \frac{1}{N} \sum_{i=1}^N CE\left(q(\mathbf{x}_i; \theta), \mathbf{y}_i\right),
$$
$$
\mathcal{L}_{BNS} = \frac{1}{L} \sum_{l=1}^L \left\| \boldsymbol{\mu}^l(\theta) - \boldsymbol{\mu}^l(\theta, \{\mathbf{x}_i\}_{i=1}^N) \right\|_2^2 + \left\| \boldsymbol{\sigma}^l(\theta) - \boldsymbol{\sigma}^l(\theta, \{\mathbf{x}_i\}_{i=1}^N) \right\|_2^2,
\tag{1}
$$

where the $l$th batch normalization (BN) layer of the pre-trained model with parameters $\theta$ (out of $L$ BN layers) stores the running mean $\boldsymbol{\mu}^l(\theta)$ and standard deviation $\boldsymbol{\sigma}^l(\theta)$ of the training dataset. The mean $\boldsymbol{\mu}^l(\theta, \{\mathbf{x}_i\}_{i=1}^N)$ and standard deviation $\boldsymbol{\sigma}^l(\theta, \{\mathbf{x}_i\}_{i=1}^N)$ are calculated on $\{\mathbf{x}_i\}_{i=1}^N$ using $\theta$. $q(\cdot; \theta)$ denotes the probability distribution by parameters $\theta$ and $CE(\cdot, \cdot)$ stands for cross-entropy loss.

The goal of the second step is to obtain the quantized model with parameters $\theta^q$, using the pre-trained model with parameters $\theta$ and synthetic dataset $\{\mathbf{x}_i\}_{i=1}^N$ with labels $\{\mathbf{y}_i\}_{i=1}^N$. We first quantize the pre-trained model with Rounding-To-Nearest (RTN) (Gupta et al., 2015), then fine-tune the quantized model with parameters $\theta^q$ with the synthetic dataset from the previous step. For strong performance, we train the quantized model by minimizing two losses, cross-entropy loss $CE(\cdot, \cdot)$ with hard label $\mathbf{y}_i$ and KL divergence loss $KL(\cdot||\cdot)$ which transfers knowledge from the pre-trained model. Note that we directly update the quantized model, while inferencing with its dequantized parameters. Equation (2) incorporates the two loss functions with balancing hyperparameter $\lambda_{CE}$.

$$
\min_{\theta^q} \mathcal{L}_{ZSQ} = \min_{\theta^q} \frac{1}{N} \sum_{i=1}^N KL\left(q(\mathbf{x}_i; \theta) || q(\mathbf{x}_i; \theta^q)\right) + \lambda_{CE} CE\left(q(\mathbf{x}_i; \theta^q), \mathbf{y}_i\right).
\tag{2}
$$

### 2.2 DIFFICULTY OF AN IMAGE

Difficulty of an image represents how easily the model $\theta$ can misclassify the image $\mathbf{x}_i$. Among various methods (Ribeiro et al., 2016; Lin et al., 2017; Kishida & Nakayama, 2019; Scheidegger et al., 2021) to evaluate the difficulty of the model in correctly classifying an image, we follow the probability-based approach (Li et al., 2019) which is used in previous ZSQ methods (Li et al., 2023a). The difficulty $\delta(\mathbf{x}_i, \theta)$ is determined by how low the model's predicted probability is for the correct label as described in Equation (3).

$$
\delta(\mathbf{x}_i, \theta) = 1 - q_{\mathbf{y}_i}(\mathbf{x}_i; \theta),
\tag{3}
$$

where $q_{\mathbf{y}_i}(\mathbf{x}_i; \theta)$ is the probability of label $\mathbf{y}_i$ predicted by the model with parameters $\theta$. This definition employs the true label to specifically highlight the model's ambiguity toward the image. Models display an error rate of 0 for difficulties below 0.5 and an increasing error rate for higher difficulties as shown in Figure 3, indicating either incorrectness or uncertainty in model predictions.

### 2.3 PROBLEM DEFINITION

Given a pre-trained image classification model and quantization bits, Zero-shot Quantization (ZSQ) targets to optimize the quantized model to maintain performance without any real images. We give the formal definition as in Problem 1.

**Problem 1** (Zero-shot Quantization). *We have a pre-trained model with parameters $\theta$ and quantization bits $B$. Zero-shot quantization is to optimize the quantized model with parameters $\theta^q$ for maximum accuracy within the $B$bit limit without the use of real data.*

## 3 OBSERVATION

We present the observations that highlight the three major challenges posed to existing methods.

**Noise in the synthetic dataset.** The synthetic dataset is noisy, as it is produced by noise optimization that starts with a Gaussian noise. In Figure 1, we compare (a) real images from the ImageNet dataset with (b) generated samples from the synthetic dataset following TexQ (Chen et al., 2023). Generated samples display distinct grainy noise that leads to an evenly distributed frequency magnitude spectrum, in contrast to the real images whose magnitude is primarily concentrated in the low-frequency area. Note that we investigate the frequency magnitude spectrum by applying Fourier transform (Cooley & Tukey, 1965; Park et al., 2021; 2024c) on images. Moreover, Figure 5 shows the severe differences in amplitude distributions between (a) real images and (b) generated samples (refer to Appendix C.3 for results on other baselines and datasets). This frequency domain discrepancy challenges the quantized model to restore classification performance during fine-tuning.

**Predictions based on off-target patterns.** Fine-tuning with a synthetic dataset leads the quantized model to rely on incorrect image patterns for predictions. Figure 2 shows the discriminative regions that Grad-CAM (Selvaraju et al., 2017) identifies for the ground-truth class across three models: (a) pre-trained ResNet-18 model on the ImageNet dataset, (b) 3bit quantized model by TexQ (Chen et al., 2023), and (c) 3bit quantized model by our SYNQ. Note that TexQ predicts based on wrong regions, unlike the pre-trained model which accurately captures critical regions (refer to Appendix C.4 for further analysis). This mismatch definitely harms the quantization performance.

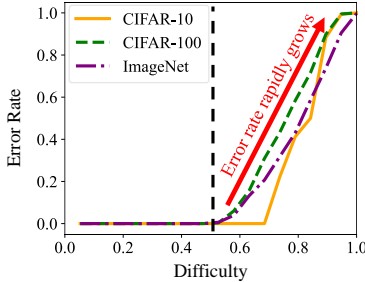

**Misguidance by erroneous hard labels.** Reliance on erroneous hard labels in the synthetic dataset leads to misguided fine-tuning outcomes. Figure 3 shows the growing error rates for pre-trained ResNet (He et al., 2016) models on CIFAR-10, CIFAR-100, and ImageNet datasets as image difficulty increases. Difficulty of an image is defined as Equation (3), detailed in Section 2.2. Consequently, the pre-trained model often mislabels samples with a difficulty level over 0.5. These erroneous hard labels of difficult samples damage quantization performance.

Figure 3: Error rates of pre-trained ResNet-20 on CIFAR-10 (yellow) and CIFAR-100 (green), and ResNet-18 on ImageNet (purple) by difficulty. Error rate rapidly grows as the difficulty exceeds 0.5.

## 4 PROPOSED METHOD

### 4.1 OVERVIEW

We propose SYNQ (**Syn**thesis-aware Fine-tuning for Zero-shot **Q**uantization), an accurate Zero-shot Quantization (ZSQ) method addressing the following three major challenges of existing methods that fine-tune with the synthetic dataset. These are the three main challenges that must be tackled:

**C1. Noise in the synthetic dataset.** Previous methods fine-tune the quantized model with a noisy synthetic dataset, which exhibits a distribution discrepancy of frequency domain compared to real images. How can we minimize the effect of the noise within the generated samples?

**C2. Prediction based on off-target patterns.** The quantized model predicts based on incorrect image regions that are unlike those observed in the pre-trained model. How can we optimize the quantized model to more accurately utilize on-target patterns?

**C3. Misguidance from erroneous hard labels.** Despite the high error rate of difficult samples, existing works trust erroneous hard labels. How can we address the misguidance by hard labels?

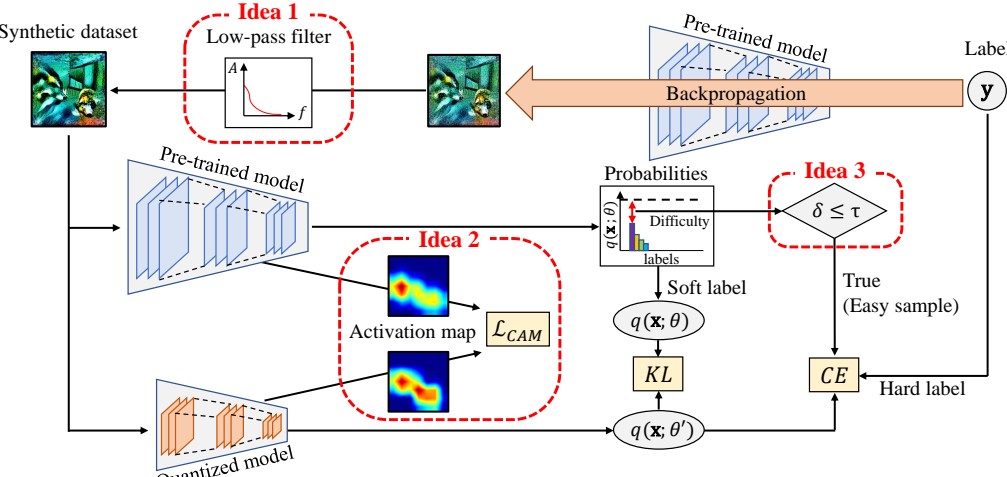

Figure 4: Overall architecture of SYNQ. Our main ideas are 1) low-pass filter, 2) alignment of class activation map, and 3) soft labels for difficult samples. See Section 4 for details.

We address these challenges with the following main ideas:

**I1.** **Low-pass filter (Section 4.2).** We directly reduce the noise from the dataset by exploiting a Gaussian low-pass filter in the frequency domain.

**I2.** **Alignment of class activation map (Section 4.3).** We align the class activation map between the pre-trained and quantized models, directly distilling knowledge to identify the correct image region from the pre-trained model to the quantized model.

**I3.** **Soft labels for difficult samples (Section 4.4).** For difficult samples, we fine-tune only with soft labels or predictions from the pre-trained model to reduce ambiguity.

Figure 4 illustrates the overall process of SYNQ. SYNQ first generates a synthetic dataset from arbitrary labels. Then, SYNQ exploits a Gaussian low-pass filter to refine the samples by removing noise. With this filtered dataset, SYNQ fine-tunes the quantized model with KL divergence and cross-entropy losses, following the standard ZSQ framework. SYNQ also optimizes CAM alignment loss $\mathcal{L}_{CAM}$ to enhance activation map alignment for better critical region detection. SYNQ exploits the threshold $\tau$ to decide on the application of cross-entropy loss based on the difficulty of a sample.

## 4.2 LOW-PASS FILTER

The first step of Zero-shot Quantization (ZSQ) is to produce the synthetic dataset which effectively mimics the real dataset. Existing methods leverage the prediction and batch normalization statistics of the pre-trained model to generate samples. However, their limitation is the noise in the synthetic dataset, as discussed in Figure 1. We investigate the intensity of this noise, by performing the Fourier transform on the datasets. Figure 5 illustrates the amplitude distribution of (a) ImageNet dataset, (b) the synthetic dataset by TexQ (Chen et al., 2023), and (c) Gaussian-filtered samples based on the distance from the center. The dark solid line indicates the mean distribution, and the surrounding colored region shows the standard deviation within a batch of 256 randomly chosen images. While the ImageNet dataset primarily exhibits lower frequency components, the synthetic dataset contains more high-frequency components, clearly indicating a higher level of sharpness and noise. This noise is observed in various ZSQ methods, regardless of the setting (see Appendix C.3).

To mitigate this noise, we exploit a Gaussian low-pass filter on the generated samples. Given a sample $\mathbf{x}_i$ with width $W$, height $H$, and filtering hyperparameter $D_0$ which is related to cut-off frequency, we compute the filtered sample $\mathbf{x}_i^F$ as shown in Equation (4).

$$\mathbf{x}_i^F = \mathcal{F}^{-1}\left(\mathbf{G} \odot \mathcal{F}(\mathbf{x}_i)\right), \mathbf{G}_{uv} = \exp\left(-\frac{(D(u,v))^2}{2D_0^2}\right), D(u,v) = \sqrt{(u - \frac{W}{2})^2 + (v - \frac{H}{2})^2}, \quad (4)$$

where $D(u,v)$ denotes the distance from the coordinate $(u,v)$ to the center in the frequency domain and $\odot$ is an element-wise multiplication. This Gaussian low-pass filter $\mathbf{G}$ works in the frequency

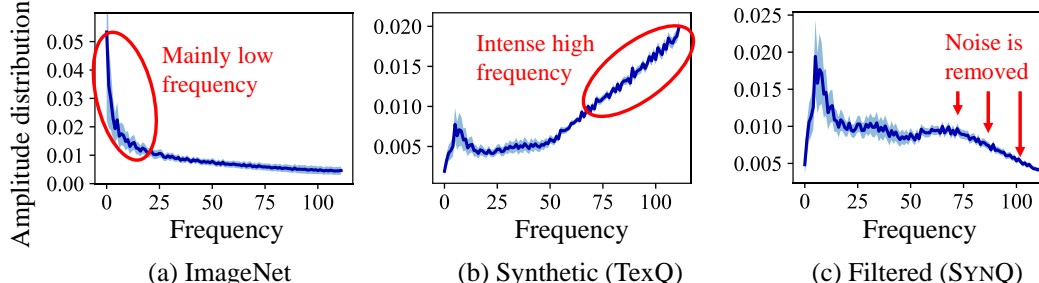

Figure 5: Comparison of amplitude distribution among (a) ImageNet dataset, (b) synthetic dataset by TexQ, and (c) filtered samples. After filtering, the distribution closely aligns with that of real images.

domain from conducting Fourier transform $\mathcal{F}$, we then apply inverse Fourier transform $\mathcal{F}^{-1}$ to obtain the filtered sample $\mathbf{x}_i^F$. Figure 5(c) clearly shows the positive effect of the filter: removal of noise in the high-frequency region resulting in an amplitude distribution aligning with that of real images. We further investigate the robustness of low-pass filter towards different types of noise in Appendix C.8.

## 4.3 ALIGNMENT OF CLASS ACTIVATION MAP

The next step is to fine-tune the quantized model to achieve high accuracy. The first challenge of this step is to ensure that the quantized model makes predictions using on-target image patterns. Existing methods fine-tune the model with classification and knowledge distillation from the pre-trained model, using the synthetic dataset. However, the quantized model from these methods fail to properly localize the object as depicted in Figure 2. To ensure the quantized model to make prediction based on correct image regions, we directly align the class activation map between pre-trained and quantized models. We optimize the class activation map (CAM) alignment loss $\mathcal{L}_{CAM}$ by minimizing the mean square error between the saliency maps $\mathbf{S}^{\theta}(\mathbf{x}_i)$ and $\mathbf{S}^{\theta^q}(\mathbf{x}_i)$ of the pre-trained and quantized models, respectively. Among various techniques (Zhou et al., 2016; Selvaraju et al., 2017; Zagoruyko & Komodakis, 2017; Chattopadhay et al., 2018) to highlight the important region of the image, we select Grad-CAM (Selvaraju et al., 2017) due to its simplicity and superiority, which we discuss further in Section 5.4. Grad-CAM generates the saliency map $\mathbf{S}^{\theta}(\mathbf{x}_i)$ by weighting the activations with their gradients, emphasizing the regions in the input image that have the greatest impact on the model's prediction. We formulate CAM alignment loss as Equation (5).

$$\mathcal{L}_{CAM}(\mathbf{x}_i; \theta, \theta^q) = \|\mathbf{S}^{\theta}(\mathbf{x}_i) - \mathbf{S}^{\theta^q}(\mathbf{x}_i)\|_F^2,$$
$$\mathbf{S}^{\theta}(\mathbf{x}_i) = \text{ReLU}\left(\sum_k \left(\frac{1}{W_k H_k} \sum_{w=1}^{W_k} \sum_{h=1}^{H_k} \frac{\partial y^{\mathbf{y}_i}}{\partial \mathbf{A}_{wh}^{k;\theta}(\mathbf{x}_i)}\right) \mathbf{A}^{k;\theta}(\mathbf{x}_i)\right), \tag{5}$$

where $\mathbf{A}^{k;\theta}(\mathbf{x}_i)$ denotes the activations of the last layer at channel $k$ with $W_k$ and $H_k$ representing its width and height, respectively. The gradient of the predicted score $y^{\mathbf{y}_i}$ for the true class $\mathbf{y}_i$ with respect to the activation $\mathbf{A}_{wh}^{k;\theta}(\mathbf{x}_i)$ at spatial location $(w, h)$ indicates the contribution of each activation to the model's prediction. Figure 2 clearly shows that $\mathcal{L}_{CAM}$ enables SYNQ to accurately target the correct image regions as the pre-trained model does (see Appendix C.4 for further analysis).

## 4.4 SOFT LABELS FOR DIFFICULT SAMPLES

The second challenge of the fine-tuning step is the misguidance from possibly mislabeled samples. Existing works assign random classes as labels for generated samples, then minimize the Inception Loss (IL) $\mathcal{L}_{IL}$ in Equation (1) to optimize the image so that the pre-trained model predicts the assigned labels. However, the pre-trained model frequently mislabels difficult samples in this approach, as the higher difficulty indicates that the pre-trained model assigns lower probabilities to the true label, following the definition in Equation (3).

To avoid this misguidance, we exclude the cross-entropy loss with the hard labels for difficult samples. We classify samples as easy or difficult based on a difficulty threshold $\tau$. For easy samples, we optimize both the cross-entropy loss with hard labels and the KL divergence with soft labels. In contrast, for difficult samples, we exclusively optimize the KL divergence with soft labels, completely omitting

the cross-entropy loss. This approach focuses on replicating the pre-trained model's responses to ambiguous images, minimizing performance degradation caused by overconfidence in hard labels. Note that previous methods apply both soft and hard labels irrespective of sample difficulty.

## 4.5 OBJECTIVE FUNCTION

Combining all three ideas of SYNQ, we modify the loss function for the fine-tuning phase from $\mathcal{L}_{ZSQ}$ in Equation (2) to $\mathcal{L}_{\text{SYNQ}}$ in Equation (6).

$$\mathcal{L}_{\text{SYNQ}} = \frac{1}{N}\sum_{i=1}^{N}\Big(KL(q(\mathbf{x}_i^F;\theta)||q(\mathbf{x}_i^F;\theta^q)) + \mathbf{1}_{\{\delta(\mathbf{x}_i^F,\theta)\leq\tau\}}\lambda_{CE}CE\big(q(\mathbf{x}_i^F;\theta^q),\mathbf{y}_i\big) + \lambda_{CAM}\mathcal{L}_{CAM}\big(\mathbf{x}_i^F;\theta,\theta^q\big)\Big), \quad (6)$$

where $\lambda_{CE}$ and $\lambda_{CAM}$ are balancing hyperparameters for cross-entropy loss and CAM alignment loss, respectively. $\mathbf{1}_{\{\cdot\}}$ is the indicator function which returns 1 if the inner statement is true and 0 otherwise. We train with the filtered samples $\mathbf{x}_i^F$ (Section 4.2) to remove the noise in the synthetic dataset. Then, we align the class activation map between the pre-trained and quantized models by optimizing $\mathcal{L}_{CAM}$ (Section 4.3) to transfer the knowledge of finding an object on the image. We also exclude cross-entropy loss for difficult samples with a threshold $\tau$ (Section 4.4) to mitigate the impact of misguidance from hard labels.

SYNQ is compatible with any ZSQ method utilizing synthetic datasets (Zhong et al., 2022b; Qian et al., 2023b; Jeon et al., 2023b). We adopt calibration center synthesis (Chen et al., 2023), difficult sample generation, and sample difficulty promotion (Li et al., 2023a) because we observe they generally perform better in ZSQ (refer to Appendix D for details). We visualize the generated images within synthetic dataset in Figure 12. The adaptability of SYNQ is clearly demonstrated through further experiments on other Zero-shot QAT and PTQ methods in Appendices C.6 and C.7, respectively. We formulate the overall algorithm of SYNQ in Algorithm 1.

**Complexity Analysis.** We analyze the time complexity of SYNQ, where $N$ and $L$ represent the numbers of training samples and layers, respectively.

**Theorem 1** (Time Complexity of SYNQ). *Given a model with an inference complexity of $O(T_\theta)$, the time complexity for the quantization procedure (Algorithm 1) of* SYNQ *is $O\big(NLT_\theta\big)$.*

*Proof.* See Appendix C.1. □

Theorem 1 demonstrates that SYNQ is an efficient approach, with a time complexity scaling linearly with the number of training samples $N$ and model layers $L$. Furthermore, SYNQ generates only 5,120 samples, making it significantly faster than generator-based methods such as AdaSG (Qian et al., 2023b) and AdaDFQ (Qian et al., 2023a), which produce over 1 million samples (see Appendix C.9 for experiments with different sizes of dataset). We perform a runtime analysis of SYNQ in Appendix C.2 to analyze the computational overhead of SYNQ.

## 5 EXPERIMENTS

We perform experiments to answer the following questions about SYNQ. Further discussions and experiments on SYNQ are discussed in Appendix C.

**Q1. Accuracy in Convolutional Neural Network (CNN) Quantization (Section 5.2).** How accurate is the quantized CNN model from SYNQ compared to those from existing ZSQ methods?

**Q2. Accuracy in Vision Transformer (ViT) Quantization (Section 5.3).** How effective is SYNQ in enhancing ViT Quantization performance?

**Q3. Analysis on Class Activation Map Techniques (Section 5.4).** Which CAM technique demonstrates the highest performance?

**Q4. Ablation Study (Section 5.5).** Are all components of SYNQ effective for enhancing the classification accuracy of the quantized model?

**Q5. Hyperparameter Analysis (Section 5.6).** How robust are the performance gains by SYNQ in hyperparameters $\lambda_{CE}$, $\lambda_{CAM}$, $D_0$, and $\tau$?

## 5.1 EXPERIMENTAL SETUP

We briefly introduce the experimental setup. Further setups are detailed in Appendix D.

Table 1: Zero-shot Quantization accuracy [%] of ResNet-20 (R-20) on CIFAR-10 and CIFAR-100, and ResNet-18 (R-20), ResNet-50 (R-50), and MobileNetV2 (MV2) on ImageNet. WBAB indicates that both weights and activations are quantized to Bbit. Note that SYNQ achieves the highest accuracy.

| Method | R-20 (CIFAR-10) | | R-20 (CIFAR-100) | | R-18 (ImageNet) | | R-50 (ImageNet) | | MV2 (ImageNet) | |
|---|---|---|---|---|---|---|---|---|---|---|
| | W4A4 | W3A3 | W4A4 | W3A3 | W4A4 | W3A3 | W4A4 | W3A3 | W4A4 | W3A3 |
| Full Precision (W32A32) | 93.89 | | 70.33 | | 71.47 | | 77.73 | | 73.03 | |
| GDFQ (Xu et al., 2020) | 90.11 | 75.11 | 63.75 | 47.61 | 60.60 | 20.23 | 54.16 | 0.31 | 59.43 | 1.46 |
| ARC (Zhu et al., 2021) | 88.55 | - | 62.76 | 40.15 | 61.32 | 23.37 | 64.37 | 1.63 | 60.13 | 14.30 |
| Qimera (Choi et al., 2021) | 91.26 | 74.43 | 65.10 | 46.13 | 63.84 | 1.17 | 66.25 | - | 61.62 | - |
| ARC + AIT (Choi et al., 2022) | 90.49 | - | 61.05 | 41.34 | 65.73 | - | 68.27 | - | 66.47 | - |
| IntraQ (Zhong et al., 2022b) | 91.49 | 77.07 | 64.98 | 48.25 | 66.47 | 45.51 | - | - | 65.10 | - |
| AdaSG (Qian et al., 2023b) | 92.10 | 84.14 | 66.42 | 52.76 | 66.50 | 37.04 | 68.58 | 16.98 | 65.15 | 26.90 |
| AdaDFQ (Qian et al., 2023a) | 92.31 | 84.89 | 66.81 | 52.74 | 66.53 | 38.10 | 68.38 | 17.63 | 65.41 | 28.99 |
| HAST (Li et al., 2023a) | 92.36 | 86.34 | 66.68 | 55.67 | 66.91 | 42.58 | - | - | 65.60 | - |
| TexQ (Chen et al., 2023) | _92.68_ | 86.47 | _67.18_ | 55.87 | _67.73_ | _50.28_ | _70.72_ | _25.27_ | _67.07_ | _32.80_ |
| PLF (Fan et al., 2024) | 92.47 | _88.04_ | 66.94 | _57.03_ | 67.02 | - | 68.97 | - | - | - |
| **SYNQ (Proposed)** | **92.76** | **88.11** | **67.34** | **57.28** | **67.90** | **52.02** | **71.05** | **26.89** | **67.27** | **34.21** |
| Standard Deviation | ± 0.10 | ± 0.15 | ± 0.15 | ± 0.29 | ± 0.19 | ± 0.34 | ± 0.17 | ± 0.24 | ± 0.21 | ± 0.27 |

**Setup.** We evaluate our method across three datasets by reporting the top-1 accuracy for the validation sets of CIFAR-10, CIFAR-100 (Krizhevsky et al., 2009) and ImageNet (ILSVRC 2012) (Deng et al., 2009) datasets. We select ResNet-20 (He et al., 2016) model for CIFAR-10 and CIFAR-100, and ResNet-18, ResNet-50 (He et al., 2016), and MobileNetV2 (Sandler et al., 2018) model for ImageNet. We follow this prevalent experimental setup from existing works (Chen et al., 2023; Qian et al., 2023a;b) to correctly compare the performance of SYNQ.

**Competitors.** We compare SYNQ with existing ZSQ methods utilizing synthetic dataset, including GDFQ (Xu et al., 2020), ARC (Zhu et al., 2021), Qimera (Choi et al., 2021), ARC + AIT (Choi et al., 2022), IntraQ (Zhong et al., 2022b), AdaSG (Qian et al., 2023b), AdaDFQ (Qian et al., 2023a), HAST (Li et al., 2023a), TexQ (Chen et al., 2023), and PLF (Fan et al., 2024). Both model weights and activation are quantized identically for all layers.

**Implementation Details.** We follow the settings from IntraQ (Zhong et al., 2022b) and HAST (Li et al., 2023a) for equal comparison. We generate 5,120 images with a batch size of 256. The batch size for fine-tuning is 256 for CIFAR-10/100 and 16 for ImageNet with epochs uniformly set to 100. We search $\tau$, $D_0$, $\lambda_{CE}$, and $\lambda_{CAM}$ within the ranges {0.5, 0.55, 0.6, 0.65, 0.7}, {20, 40, 60, 80, 100}, {0.005, 0.05, 0.5, 5}, and {20, 50, 100, 200, 300, 500, 2000}, respectively. All of our experiments were done at a workstation with Intel Xeon Silver 4214 and RTX 3090.

## 5.2 ACCURACY IN CNN QUANTIZATION (Q1)

We evaluate the quantization accuracy of SYNQ against existing ZSQ methods using CIFAR-10, CIFAR-100, and ImageNet datasets. Our method significantly enhances quantized model accuracy on all settings with 3bit and 4bit quantization as summarized in Table 1. We report the mean and standard deviation of 5 iterations, each using different random seed values. We have two observations from the result. First, SYNQ benefits the fine-tuning of quantized models consistently across diverse quantization bits, models, and datasets. Compared to state-of-the-art methods TexQ (Chen et al., 2023) and PLF (Fan et al., 2024), SYNQ achieves higher accuracies of up to 1.74%p (ResNet-18 on ImageNet dataset). Second, SYNQ demonstrates increasing effectiveness as bit-width decreases. Considering that lower-bit quantization is inherently more challenging, our results clearly showcase the robustness of SYNQ due to its effective fine-tuning that overcomes the aforementioned limitations.

## 5.3 ACCURACY IN VIT QUANTIZATION (Q2)

We investigate the effectiveness of SYNQ in enhancing ZSQ performance for Vision Transformers (ViTs). Table 2 shows the ZSQ precision of four ViT models, DeiT-Tiny, DeiT-Small (Touvron et al., 2021), Swin-Tiny, and Swin-Small (Liu et al., 2021b) pre-trained on ImageNet dataset. SYNQ enhances the quantization precision across various models, achieving up to 0.58%p increase in

Table 2: Zero-shot Quantization accuracy [%] of ViT models on ImageNet dataset. WBAB indicates that both weights and activations are quantized to Bbit. Note that SYNQ shows consistent improvements in quantization performance across various models.

| Bits | Method | DeiT-Tiny | DeiT-Small | Swin-Tiny | Swin-Small | Average |
|---|---|---|---|---|---|---|
| | Full Precision | 72.21 | 79.85 | 81.35 | 83.20 | 79.15 |
| W4A8 | PSAQ-ViT (Li et al., 2022) | $65.57 \pm 0.10$ | $72.04 \pm 0.19$ | $69.78 \pm 1.67$ | $75.03 \pm 0.63$ | 70.61 |
| | **SYNQ (Proposed)** | $\mathbf{65.90 \pm 0.07}$ | $\mathbf{72.28 \pm 0.34}$ | $\mathbf{70.76 \pm 1.61}$ | $\mathbf{75.82 \pm 0.54}$ | **71.19** |
| W8A8 | PSAQ-ViT (Li et al., 2022) | $71.56 \pm 0.03$ | $75.97 \pm 0.20$ | $73.54 \pm 1.61$ | $76.68 \pm 0.53$ | 74.44 |
| | **SYNQ (Proposed)** | $\mathbf{71.74 \pm 0.03}$ | $\mathbf{76.16 \pm 0.29}$ | $\mathbf{74.11 \pm 1.82}$ | $\mathbf{77.32 \pm 0.59}$ | **74.83** |

average precision when applied to the recent method PSAQ-ViT (Li et al., 2022). The results show that SYNQ is an accurate ZSQ method not only tailored for CNN but also is effective in ViTs.

## 5.4 ANALYSIS ON CLASS ACTIVATION MAP TECHNIQUES (Q3)

We compare the quantization accuracy of SYNQ when utilizing different techniques to output the class activation map. We show the 3bit quantization accuracy of ResNet-18 model in Figure 6. Grad-CAM (Selvaraju et al., 2017) demonstrates higher performance over CAM (Zhou et al., 2016) and Grad-CAM++ (Chattopadhay et al., 2018). This is attributed to Grad-CAM++ being specialized in localizing multiple objects, whereas Grad-CAM focuses on a single object. Additionally, note that Grad-CAM also takes advantage over CAM in that it is a direct generalization of CAM which is applicable only to models with a global pooling layer. Thus, we utilize Grad-CAM to generate the saliency map for the CAM alignment loss $\mathcal{L}_{CAM}$, as described in Section 4.4.

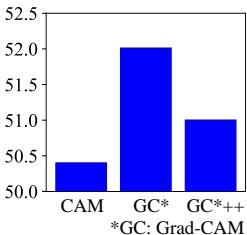

Figure 6: ZSQ accuracy comparison on different CAM techniques. See Section 5.3 for details.

## 5.5 ABLATION STUDY (Q4)

We perform an ablation study to show that each main idea of SYNQ, such as low-pass filter (I1) in Section 4.2, alignment of class activation map (I2) in Section 4.3, and soft labels for difficult samples (I3) in Section 4.4, improves the classification accuracy of the compressed model. We summarize the 3bit quantization results of ResNet-18 model on ImageNet dataset in Table 3. Note that the baseline denotes HAST (Li et al., 2023a) with layer-wise batch normalization loss from TexQ (Chen et al., 2023) as detailed in Appendix D. Our analysis shows that all proposed ideas contribute to improved performance, with low-pass filter (I1) having the strongest impact of 5.80%p.

Table 3: Ablation study on the main ideas of SYNQ. All ideas contribute to the improved performance.

| I1 | I2 | I3 | Accuracy [%] |
|---|---|---|---|
| Baseline | | | 43.63 |
| ✓ | | | 49.43 |
| | ✓ | | 48.26 |
| | | ✓ | 46.42 |
| ✓ | ✓ | | 51.24 |
| ✓ | | ✓ | 50.81 |
| | ✓ | ✓ | 50.06 |
| ✓ | ✓ | ✓ | **52.02** |

## 5.6 HYPERPARAMETER ANALYSIS (Q5)

We analyze the robustness of SYNQ concerning the newly introduced hyperparameters $\lambda_{CE}$, $\lambda_{CAM}$, $D_0$, and $\tau$ in Figure 7. We report the 3bit quantization accuracy for the ResNet-18 model trained on the ImageNet dataset. We have three observations from the result. First, as shown in Figure 7(a), the classification accuracy remains robust across a range of $\lambda_{CE}$ and $\lambda_{CAM}$ values. This robustness indicates that SYNQ remains effective even when these hyperparameters are not precisely tuned. Second, Figure 7(b) illustrates the effect of varying the difficulty threshold $\tau$. Note that the classification accuracy increases as $\tau$ increases from 0 to 0.5, since too low $\tau$ excludes many useful samples for cross-entropy training. However, the classification accuracy starts to decrease as $\tau$ becomes greater than 0.5, since it allows to use difficult and ambiguous samples for cross-entropy training. We observe that the $\tau$ value of 0.5 gives the best trade-off between using more samples and not using ambiguous samples. We further conduct a deeper analysis on $\tau$ in Appendix C.11, verifying its impact on different settings. Third, Figure 7(c) shows that an appropriate balance in $D_0$ is necessary to maintain performance. Extremely low $D_0$ values result in significant performance degradation due to excessive filtering, which oversmooths the images and results in the loss of crucial information. Overall, SYNQ consistently outperforms baselines across a diverse range of hyperparameter values.

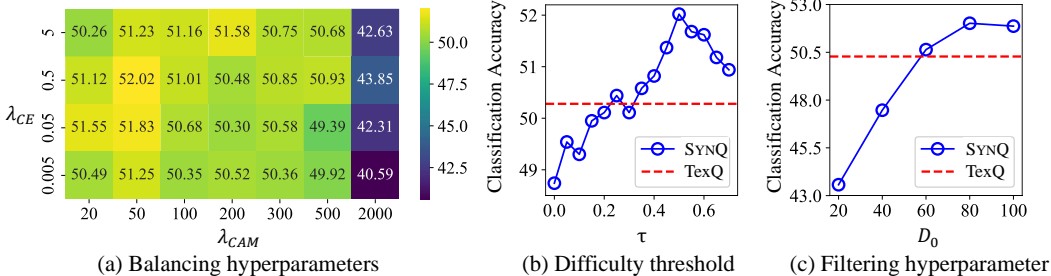

Figure 7: Hyperparameter analysis on (a) balancing hyperparameters $\lambda_{CE}$ and $\lambda_{CAM}$, (b) difficulty threshold $\tau$, and (c) filtering hyperparameter $D_0$. See Section 5.5 for details.

# 6 RELATED WORK

**Network Quantization.** Network quantization reduces the computational complexity and memory footprint of deep neural networks by converting the weights, activations, or both from full precision to lower precision formats (Deng et al., 2020; Ada; Guo et al., 2022; Shang et al., 2023; Park et al., 2024b). Quantization significantly speeds up inference and reduces power consumption, enabling deployment on edge devices with limited resources. Recent advancements in network quantization include Quantization-Aware Training (QAT) (Jacob et al., 2018; Lee et al., 2022; Dettmers et al., 2023; Xu et al., 2024) and Post-Training Quantization (PTQ) (Li et al., 2021; Frantar et al., 2023; Zhong et al., 2022a; Jeon et al., 2022). QAT integrates quantization into training, enabling the model to learn weights robust to quantization noise, thereby maintaining higher accuracy. On the other hand, PTQ quantizes pre-trained models using calibration to minimize accuracy loss without original training data. Furthermore, advanced strategies such as mixed-precision quantization (Koryakovskiy et al., 2023), knowledge distillation (Boo et al., 2021), adaptive quantization (Zhou et al., 2018), weight sharing (Ullrich et al., 2016), parameter reparameterization (Li et al., 2023c) and hardware-awareness (Wang et al., 2019) have shown promising results in achieving a balance between model efficiency and performance. However, existing works require real data to directly train or calibrate the quantized model. In contrast, SYNQ focuses on QAT scenarios where there is no access to real data.

**Zero-shot Quantization.** Zero-shot Quantization (ZSQ) (Cai et al., 2020), also called as data-free quantization (Nagel et al., 2019; Chen et al., 2019; Choi et al., 2020), performs quantization without the need for accessing the training data of full-precision models. Earlier methods focused on calibrating model parameters solely based on model properties without acquiring any data (Banner et al., 2019; Guo et al., 2022). Unfortunately, these methods resulted in significant performance drops at lower bit widths such as 3bit or 4bit quantization (Xu et al., 2020; Zhong et al., 2022b). Recent studies generate synthetic datasets and fine-tune the quantized model to enhance performance (Haroush et al., 2020; Choi et al., 2021; Liu et al., 2021a; Zhong et al., 2022b). GDFQ (Xu et al., 2020) first employs generative methods leveraging batch normalization statistics and extra category label information. Numerous variants have developed the field by introducing techniques such as advanced generators (Zhu et al., 2021), boundary supporting samples (Choi et al., 2021), noise optimization (Cai et al., 2020), diversified samples (Zhang et al., 2021), intra-class heterogeneity (Cai et al., 2020), hard sample generation (Li et al., 2023a), texture feature calibration (Chen et al., 2023), and pseudo-label filtering (Fan et al., 2024). Recently, several works further advances ZSQ into Vision Transformers (Li et al., 2022; 2023b; Ramachandran et al., 2024). However, existing methods continue to struggle with the three primary challenges (see Section 4.1). In contrast, SYNQ tackles these challenges with three main ideas: low-pass filter, class activation map alignment, and soft labels for difficult samples.

# 7 CONCLUSION

We propose **SYNQ** (**Syn**thesis-aware Fine-tuning for Zero-shot **Q**uantization), an accurate Zero-shot Quantization (ZSQ) method that effectively addresses the three major limitations of fine-tuning with synthetic datasets: 1) noise in the synthetic dataset, 2) predictions based on off-target patterns, and the 3) misguidance by erroneous hard labels. We exploit a low-pass filter to minimize noise, align the class activation map to ensure prediction from correct image region, and leverage soft labels on difficult samples to avoid misguidance by erroneous hard labels. SYNQ consistently outperforms existing ZSQ methods across diverse models, quantization bits, and datasets. Future works include extending our method into settings such as object detection and diffusion models.

## ACKNOWLEDGMENTS

This work was supported by Institute of Information & communications Technology Planning & Evaluation (IITP) grant funded by the Korea government (MSIT) [No.RS-2020-II200894, Flexible and Efficient Model Compression Method for Various Applications and Environments], [No.RS-2021-II211343, Artificial Intelligence Graduate School Program (Seoul National University)], and [No.RS-2021-II212068, Artificial Intelligence Innovation Hub (Artificial Intelligence Institute, Seoul National University)]. This work was supported by Youlchon Foundation. The Institute of Engineering Research at Seoul National University provided research facilities for this work. The ICT at Seoul National University provides research facilities for this study. U Kang is the corresponding author.

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

## A  NOTATION

We summarize the frequently used notations in the paper as Table 4.

Table 4: Frequently used notations.

| Symbol | Description |
|---|---|
| $\theta$ | A pre-trained model |
| $\theta^q$ | The quantized model |
| $\{\mathbf{x}_i\}_{i=1}^N$ | Synthetic samples |
| $\{\mathbf{y}_i\}_{i=1}^N$ | One-hot encoded labels of synthetic samples |
| $q(\mathbf{x}_i; \theta)$ | Probability distribution of a sample $\mathbf{x}_i$ predicted by parameters $\theta$ |
| $\delta(\mathbf{x}_i; \theta)$ | Difficulty of a sample $\mathbf{x}_i$ predicted by parameters $\theta$ |
| $KL(\cdot\|\|\cdot)$ | KL divergence loss |
| $CE(\cdot, \cdot)$ | Cross-entropy loss |
| $\mathcal{F}$ | Fourier transform function |
| $\mathbf{G}$ | Gaussian low-pass filter |
| $D_0$ | Filtering hyperparameter |
| $\lambda_{CAM}$ | Balancing hyperparameter of CAM loss |
| $\lambda_{CE}$ | Balancing hyperparameter of CE loss |
| $\tau$ | Threshold of difficulty for cross-entropy loss |

## B  ALGORITHM

We describe the quantization procedure of SYNQ in Algorithm 1. Note that any technique for generating synthetic datasets is applicable.

---

**Algorithm 1** Quantization procedure of SYNQ

---

**Input:** the pre-trained model with parameters $\theta$, hyperparameters $n_{ep}$, $D_0$, $\lambda_{CAM}$, $\lambda_{CE}$, and $\tau$.
**Output:** the parameters $\theta^q$ of the quantized model.
    **/** Step 1: Generate synthetic dataset **/**
1: Initialize the synthetic dataset $\{\mathbf{x}_i\}_{i=1}^N$ with Gaussian noise.
2: Randomly assign labels $\{\mathbf{y}_i\}_{i=1}^N$ for synthetic dataset.
3: Optimize $\{\mathbf{x}_i\}_{i=1}^N$ to minimize $\mathcal{L}_{IL} + \alpha\mathcal{L}_{BNS}$.      ▷ Compatible with any synthetic dataset

    **/** Step 2: Fine-tune quantized model **/**
4: Initialize $\theta^q$ following the round-to-nearest scheme.
5: Apply a low-pass Gaussian filter with a cut-off frequency $D_0$
    to synthetic samples and obtain $\{\mathbf{x}_i^F\}_{i=1}^N$.      ▷ Idea 1: Low-pass filter
6: **for** each epoch in $[1, \ldots, n_{ep}]$ **do**
7:     Initialize the total loss $\mathcal{L}$ to zero.
8:     **for** $i$ in $[1, \ldots, N]$ **do**
9:         Perform forward pass of $\theta$ and $\theta^q$ with synthetic sample $\mathbf{x}_i^F$.
10:        Compare gradients and calculate the CAM loss $\mathcal{L}_{CAM}$.      ▷ Idea 2: CAM alignment
11:        $\mathcal{L} \leftarrow \mathcal{L} + KL(q(\mathbf{x}_i; \theta)\|\|q(\mathbf{x}_i; \theta^q)) + \lambda_{CAM}\mathcal{L}_{CAM}$
12:        Calculate $\delta(\mathbf{x}_i; \theta)$
13:        **if** $\delta(\mathbf{x}_i; \theta) \leq \tau$ **then**      ▷ Idea 3: Soft labels for difficult samples
14:           $\mathcal{L} \leftarrow \mathcal{L} + \lambda_{CE}CE(q(\mathbf{x}_i; \theta^q), \mathbf{y}_i)$
15:        **end if**
16:     **end for**
17:     Update $\theta^q$ to minimize $\mathcal{L}$.
18: **end for**
19: **return** $\theta^q$

---

## C  FURTHER DISCUSSION AND EXPERIMENTS

### C.1  PROOF OF THEOREM 1

We provide the proof of Theorem 1 as below:

*Proof.* We investigate the time complexity of SYNQ in three steps: synthetic dataset generation, low-pass filter application, and quantized model fine-tuning. First, synthetic dataset generation involves calculating Inception Loss $\mathcal{L}_{IL}$ and Batch Normalization Statistics Loss $\mathcal{L}_{BNS}$ for $N$ samples, each requiring a forward pass through the model, resulting in complexity of $O(NT_\theta)$.

Second, applying the low-pass filter $\mathbf{G}$ involves a Fourier transform $\mathcal{F}(\cdot)$, an element-wise multiplication $\odot$, and an inverse Fourier transform $\mathcal{F}^{-1}(\cdot)$ (see Equation 4). Implementing with Fast Fourier Transform(FFT), the time complexity for a single input $\mathbf{x}$ with size of $Z \times Z$ is $O(Z \log Z)$, $O(Z)$, and $O(Z \log Z)$) is for $\mathcal{F}(\mathbf{x})$, $\mathbf{G} \odot \mathcal{F}(\mathbf{x})$, and $\mathcal{F}(\mathbf{G} \odot \mathcal{F}(\mathbf{x}))$, respectively (Cooley & Tukey, 1965). Therefore, the time complexity of this step is $O(NZ \log Z)$ for $N$ samples.

Lastly, the fine-tuning step involves calculating the loss for $N$ filtered samples, resulting in complexity of $O(N(T_\theta + L \cdot T_\theta))$. Here, $T_\theta$ represents the complexity of computing the cross-entropy and KL divergence losses, and $L \cdot T_\theta$ represents the complexity of computing the Grad-CAM loss $\mathcal{L}_{CAM}$ across $L$ layers. Generating saliency maps $S^\theta(\mathbf{x})$ for a single input $\mathbf{x}$ for all $L$ layers using Grad-CAM requires one forward pass and $L$ backward passes, thereby requiring $O(NLT_\theta)$ for $N$ samples (Selvaraju et al., 2017). Thus, the complexity of computing $\mathcal{L}_{CAM}$ (Equation 5) is $O(NLT_\theta)$ because aligning the saliency maps $\mathbf{S}^\theta(\mathbf{x}_i)$ and $\mathbf{S}^{\theta^q}(\mathbf{x}_i)$ is negligible compared to model inference time $T_\theta$. In summary, this step is simplified to $O(NLT_\theta)$.

Combining these complexities, we get:

$$O(NT_\theta + NZ \log Z + NLT_\theta) = O(NLT_\theta) \quad (\because T_\theta \gg Z \log Z).$$

$\square$

### C.2  RUNTIME ANALYSIS

We perform a runtime analysis to investigate the computational overhead introduced by SYNQ. For this, we measure the difference in fine-tuning time between the baseline methods with and without SYNQ. For fair comparison, we compare only with noise optimization methods since they do not train a generator model while fine-tuning. Figure 8 depicts the relative contribution of baseline methods to the per-epoch fine-tuning time compared to the approach where SYNQ is added to the baseline method for three baselines, IntraQ (Zhong et al., 2022b), HAST (Li et al., 2023a), and TexQ (Chen et al., 2023). Note that the overhead from SYNQ is marginal, i.e., the added time takes only 17.81% of the total time in average. Thus, SYNQ improves adopted models with minimal sacrifice of quantization time.

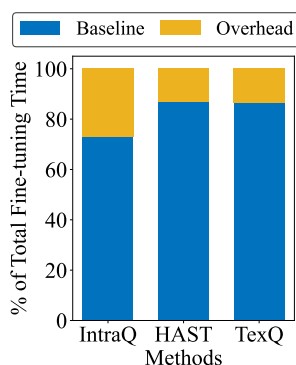

Figure 8: Runtime analysis of SYNQ. The overhead from SYNQ is marginal.

### C.3  PREVALENT NOISE IN THE SYNTHETIC DATASET

In Section 4.2 and Figure 5, we empirically analyze the limitation of the exiting ZSQ approaches, i.e., their synthetic datasets contain more high-frequency components compared to real image datasets, clearly indicating a higher level of noise. We investigate this limitation across various ZSQ methods and datasets to demonstrate that it is a widespread issue, not confined to specific scenarios. Figures 9 and 10 show the amplitude distribution of real and synthetic datasets, respectively. We compare three real datasets CIFAR-10, CIFAR-100, and ImageNet with the corresponding synthetic dataset produced by three baseline methods, IntraQ (Zhong et al., 2022b) (first row), HAST (Li et al., 2023a) (third row), and TexQ (Chen et al., 2023) (fifth row). We then apply a low-pass filter ($D_0 = 50$) to mitigate the observed noise. As shown in the figures, the discrepancy in amplitude distribution is observed regardless of the baseline method or dataset. This is effectively mitigated by exploiting the filter that removes high-frequency noise, thereby leading to enhanced quantization performance. In summary, both the limitation of noise in synthetic dataset and effect of low-pass filter (Section 4.2) are evident in a large variety of settings.

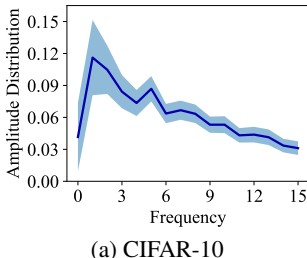 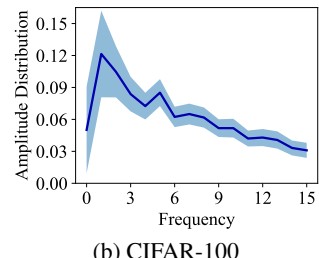 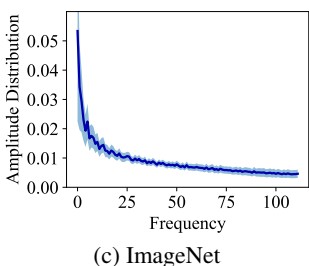

| (a) CIFAR-10 | (b) CIFAR-100 | (c) ImageNet |

Figure 9: Amplitude distribution of various real image datasets. See Appendix C.3 for details.

Table 5: The average per-batch KL-divergence [$\times 10^{-2}$] between the saliency maps of the pre-trained and quantized models trained with different methods and datasets. See Section C.4 for details.

| Method | Real dataset | Synthetic dataset | |
| --- | --- | --- | --- |
| | | Baseline | + SYNQ |
| IntraQ (Zhong et al., 2022b) | 3.1973 | 4.0251 | **3.2891 (-18.29%)** |
| HAST (Li et al., 2023a) | 3.0976 | 3.9867 | **3.3133 (-16.89%)** |
| TexQ (Chen et al., 2023) | 2.9952 | 3.8436 | **3.1542 (-17.94%)** |

## C.4 FURTHER ANALYSIS ON CAM PATTERN DISCREPANCY

The observation of "predictions based on off-target patterns" from Figure 2 in Section 3 is intuitive and persuasive, but it is analyzed under limited conditions. We explore the distance between saliency maps derived from Grad-CAM (Selvaraju et al., 2017) to 1) validate this challenge across diverse methods and 2) demonstrate that it applies not only to a few selected images but also to the entire synthetic dataset on average. Table 5 presents the average distance between the saliency maps of the pre-trained model (target) and the quantized models (prediction) trained with various methods and datasets. We compute the KL divergence between the saliency maps of 3-bit quantized ResNet-18 models pre-trained on the ImageNet dataset, treating each saliency map as a distribution, and report the average distance across batches with size of 32. From the result, we have three observations. First, all three baseline methods demonstrate notable CAM discrepancies, emphasizing the generality of this challenge in the ZSQ domain. Second, this challenge is significantly mitigated when using real datasets, with a reduction in distance exceeding 20% compared to synthetic datasets. This highlights that training with synthetic datasets exacerbates this problem. Last, adopting SYNQ significantly lowers the CAM discrepancy for all baseline methods, achieving a reduction of approximately 16-18% compared to the baseline. The resulting discrepancy is comparable to that of training with real datasets. Overall, the challenge of CAM pattern discrepancy 1) is evident across multiple methods and 2) is notably reduced by CAM alignment of SYNQ.

## C.5 COMPARISON BETWEEN CAM ALIGNMENT AND FEATURE ALIGNMENT

We compare CAM (Class Activation Map) alignment of our proposed SYNQ and feature alignment of HAST (Li et al., 2023a) to mitigate possible misunderstandings and highlight the novelty of the proposed idea. The main difference between CAM alignment and feature alignment lies in their focus on different aspects of the model's behavior. Compared to activation maps that show the response of the model to the given input, CAM emphasizes the region of the model related to the model's prediction, highlighting the most relevant features that contribute to the final decision. This is because CAM is defined based on the magnitude of the gradient with respect to the cross-entropy between the prediction and the label. Con-

Table 6: Ablation study of two alignment techniques. CAM alignment shows superior performance.

| FA | I2 | I1 & I3 | Accuracy [%] |
| --- | --- | --- | --- |
| Baseline | | | 43.63 |
| ✓ | | | $46.77 \pm 0.30$ |
| | ✓ | | $\mathbf{48.26} \pm 0.29$ |
| ✓ | | ✓ | $51.20 \pm 0.30$ |
| | ✓ | ✓ | $\mathbf{52.02} \pm 0.34$ |

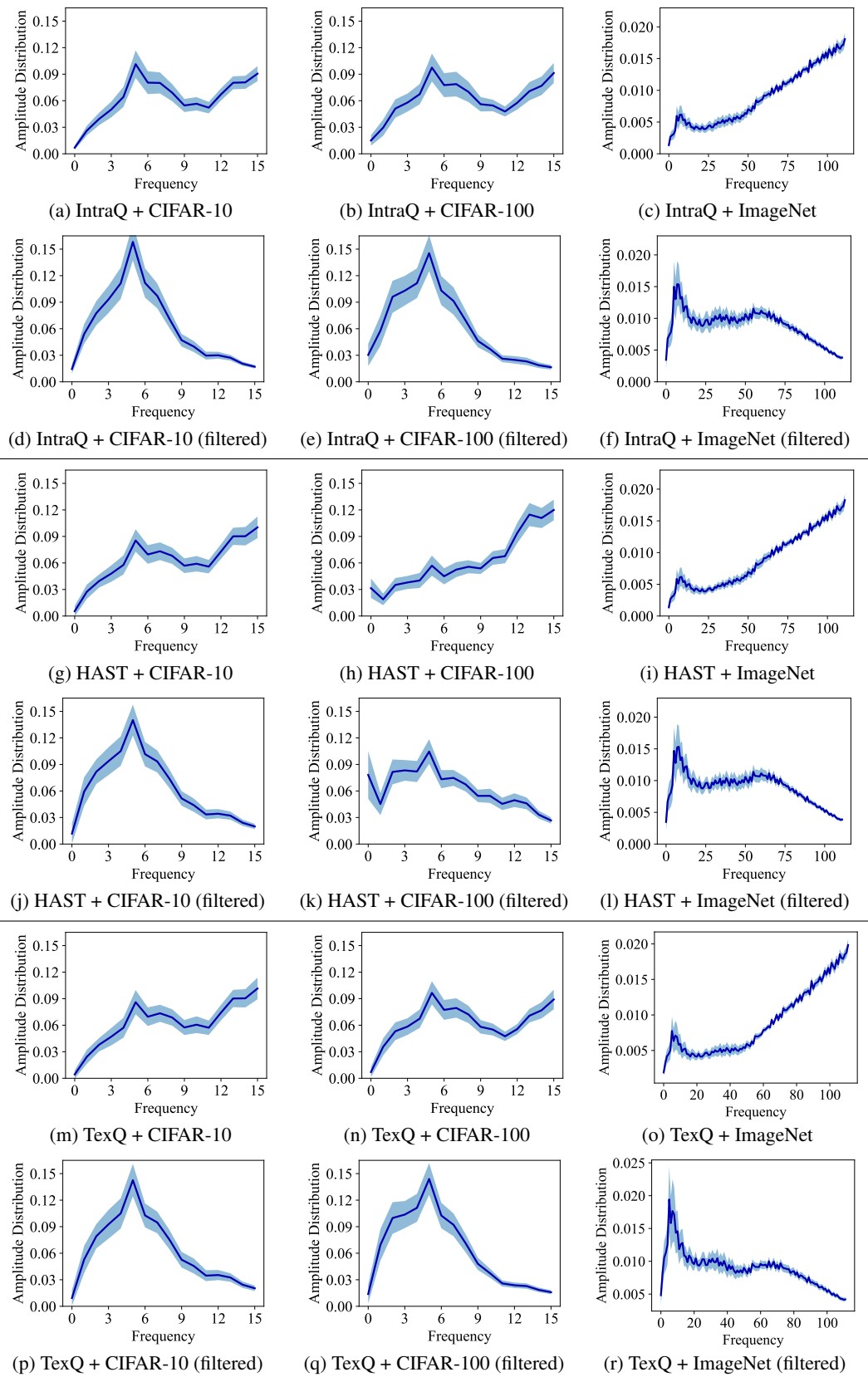

Figure 10: Amplitude distribution of various synthetic datasets. See Appendix C.3 for details.

Table 7: Comparison of ZSQ accuracy [%] of the ResNet-18 model pre-trained on the ImageNet dataset, before and after applying SYNQ. Regardless of the baseline method and quantization bits, SYNQ consistently improves the ZSQ accuracy.

| Type | Method | W3A3 | | | W4A4 | | |
|---|---|---|---|---|---|---|---|
| | | Baseline | + SYNQ | Imp. | Baseline | + SYNQ | Imp. |
| Generator-based | GDFQ (Xu et al., 2020) | 20.23 | $25.57 \pm 0.28$ | **5.34** | 60.60 | $61.23 \pm 0.21$ | **0.63** |
| | Qimera (Choi et al., 2021) | 1.17 | $32.34 \pm 0.30$ | **31.17** | 63.84 | $64.28 \pm 0.18$ | **0.44** |
| | AdaDFQ (Qian et al., 2023a) | 38.10 | $40.56 \pm 0.28$ | **2.46** | 66.53 | $66.79 \pm 0.22$ | **0.26** |
| Noise Optimization | IntraQ (Zhong et al., 2022b) | 45.51 | $50.44 \pm 0.42$ | **4.93** | 66.47 | $66.73 \pm 0.19$ | **0.26** |
| | HAST (Li et al., 2023a) | 42.58 | $50.69 \pm 0.38$ | **8.11** | 66.91 | $67.19 \pm 0.21$ | **0.28** |
| | TexQ (Chen et al., 2023) | 50.28 | $51.58 \pm 0.30$ | **1.30** | 67.73 | $67.85 \pm 0.16$ | **0.12** |

sidering that simple fine-tuning methods unintentionally lead the quantized model to rely on incorrect image patterns for predictions, CAM alignment shows clear advantage over feature alignment. By aligning the saliency maps between the original and quantized models, we ensure that the critical predictive regions remain consistent, thereby preserving the interpretability and accuracy of the model's decisions, which is more effective than merely matching activation maps.

We conduct an ablation study to compare the performance. Table 6 reports the 3bit ZSQ accuracy of ResNet-18 model on ImageNet dataset when applying CAM alignment (I2) and feature alignment (FA). CAM alignment shows clear advantage in performance over feature alignment both with and without other ideas (I1 & I3) of SYNQ.

Furthermore, we compare the computational overhead of two alignments. They have the same time complexity since both maps are obtained by applying backpropagation through the network. In practice, the average training time (in seconds) of the ResNet-18 model per epoch is $113.40 \pm 2.28$ seconds and $113.27 \pm 2.34$ seconds for CAM alignment and feature alignment, respectively. Regarding training time, the gap between two methods is negligible. Overall, CAM alignment directly targets the second challenge (prediction based on off-target patterns), thereby showing notable performance enhancement with similar training time compared to the feature alignment of HAST.

### C.6 APPLICATION ON DIFFERENT BASELINES

We evaluate the ZSQ performance when applying SYNQ on different baselines to investigate the adaptability of the proposed method. Specifically, we select three generator-based baselines (GDFQ (Xu et al., 2020), Qimera (Choi et al., 2021), and AdaDFQ (Qian et al., 2023a)) and three noise optimization baselines (IntraQ (Zhong et al., 2022b), HAST (Li et al., 2023a), and TexQ (Chen et al., 2023)). While generator-based methods simultaneously train the generator and quantized model, noise optimization methods generate the synthetic dataset first and fine-tune with it afterwards. Table 7 reports the accuracy of quantized models and the percent point improvement ("Imp."). SYNQ consistently enhances the ZSQ performance of for all baseline methods, specifically up to 31.17%p. The increasing effectiveness in lower bit-width experiments highlights the superiority of SYNQ. In overall, SYNQ is powerful and versatile since it is easily integrated with any ZSQ method utilizing synthetic dataset, enhancing their performance across various settings.

### C.7 ANALYSIS ON ZERO-SHOT POST-TRAINING QUANTIZATION SETTING

In this paper, we mainly discover ZSQ under settings that additional fine-tuning of the quantized model is performed, namely *Quantization-Aware Training (QAT)* setting. However, a recent work, Genie (Jeon et al., 2023b) has explored ZSQ under *Post-Training Quantization (PTQ)* setting, where no additional fine-tuning is needed. In this section, we first briefly discuss the preliminaries on uniform quantization. Then, we compare the settings of SYNQ and Genie to mitigate possible misunderstandings and explain why Genie is neglected from our competitors in the main experiments (Table 1). Lastly, we integrate SYNQ with Genie and evaluate its quantization performance to highlight the superior adaptability and broad applicability of SYNQ.

Table 8: Zero-shot Quantization accuracy [%] of a ResNet-18 model on ImageNet quantized with Genie (Jeon et al., 2023b) as baseline. WPAQ indicates that weights and activations are quantized each into Pbit and Qbit, respectively. SYNQ shows consistent improvements in quantization performance when applied to Genie, across various quantization bits.

| Method | W2A2 | W2A4 | W3A3 | W4A4 | Average |
|---|---|---|---|---|---|
| Genie (Jeon et al., 2023b) | 54.01 | 65.10 | 66.84 | 69.66 | 63.90 |
| + SYNQ (Proposed) | $54.97 \pm 0.35$ | $65.88 \pm 0.27$ | $67.42 \pm 0.21$ | $69.88 \pm 0.19$ | **64.54** |

**Preliminaries on Uniform Quantization.** We describe the preliminaries on the uniform quantization scheme. Uniform quantization is to represent the weight and activation of a higher-bit given network, within lower bit integers. To perform $B$bit uniform quantization, we first linearly scale the distribution of weight matrix $\mathbf{W}$ within a range of $[-2^{B-1}, 2^{B-1} - 1]$, then map weight values into equally divided integers following the rounding-to-nearest scheme (Gupta et al., 2015). Given a matrix $\mathbf{W}$ with the size of quantization granularity, the $B$bit quantized matrix $\mathbf{W}^q$ by uniform quantization is calculated as shown in Equation 7.

$$\mathbf{W}^q = \lfloor \frac{\mathbf{W}}{s} - z + \frac{1}{2} \rfloor, \quad \text{where} \quad s = \frac{\beta - \alpha}{2^B - 1} \ , \ z = \frac{\alpha}{s} + 2^{B-1}, \tag{7}$$

and $[\alpha, \beta]$ is the clipping range corresponding to $[-2^{B-1}, 2^{B-1} - 1]$ in integer scale. Properly choosing the clipping range $[\alpha, \beta]$ for $\mathbf{W}$ is essential, as it defines the scaling factor $s$ and zero-point $z$ required for accurate quantization. A commonly used approach, known as *Min-max Quantization*, involves setting $\alpha$ and $\beta$ to the minimum and maximum values of $\mathbf{W}$, respectively. The key advantage of min-max quantization is its simplicity and effectiveness, requiring no calibration to define the clipping range. This makes it the most essential and unbiased baseline for fair comparison among diverse quantization methods, especially under QAT setting.

However, min-max quantization is vulnerable to outliers, especially when quantizing activation. Outliers significantly expand the range $[\alpha, \beta]$, leading to lower precision for the majority of values in $\mathbf{W}$ during quantization. Parameterized clipping (Choi et al., 2018) mitigates this outlier effects by allowing the range to adapt based on the data distribution. This method leverages a calibration dataset to identify clipping thresholds $\alpha$ and $\beta$ that best represent the data distribution for improved quantization precision. Building on this approach, advanced techniques such as adaptive rounding (Nagel et al., 2020), learned step size (Esser et al., 2020), random dropping (Wei et al., 2022), block reconstruction (Li et al., 2021), and scale reparameterization (Li et al., 2023c) have been developed under PTQ settings, further enabling improved quantization without fine-tuning.

**A Direct Comparison with Genie (Jeon et al., 2023b).** We compare the settings between SYNQ and Genie (Jeon et al., 2023b). Whereas SYNQ optimizes the parameters $\theta^q$ of the quantized model in QAT, Genie follows a PTQ scheme, focusing on the scale factor $s$ and zero-point $z$. To achieve this, Genie combines a joint optimization framework for PTQ (Genie-M) for $s$ and soft-bit $\mathbf{V}$ (refer to AdaRound (Nagel et al., 2020) and Genie (Jeon et al., 2023b) for details) with advanced techniques such as LSQ (Esser et al., 2020), QDrop (Wei et al., 2022), and BRECQ (Li et al., 2021). Moreover, Genie fixes the quantization bits of the first layer's weights and activation, as well as the last layer's activation, to 8 bits across all experiments. On the other hand, SYNQ and other QAT approaches that are listed in Table 1 uniformly assign bits across all layers, using min-max quantization as the baseline. Due to the differences in quantization strategies and experimental conditions, evaluating Genie alongside zero-shot QAT methods is challenging.

**Accuracy in Zero-shot Post-Training Quantization.** While Genie's PTQ framework does not support experiments under min-max quantization, our proposed method SYNQ enables synthesis-aware fine-tuning for any ZSQ method that generates and utilizes synthetic datasets. Consequently, we evaluate the ZSQ accuracy under Genie's setting both with and without SYNQ in Table 8. We follow Genie for the experimental settings (see Table 3 and Appendix A of the Genie paper) and carry out implementation using their official code. Note that the size of synthetic dataset is 1,024 for this experiment. Applying SYNQ leads to consistent gains in ZSQ accuracy for Genie across various bit settings, showing an average enhancement of 0.66%p. These results clearly demonstrate the superiority of SYNQ, showcasing its compatibility with diverse quantization techniques other than min-max quantization.

Table 9: 3bit ZSQ accuracy [%] of a ResNet-18 model pre-trained on ImageNet dataset when four different types of noise is injected into its synthetic dataset. See Section C.8 for details.

| Method | Baseline | Noise | | | |
| --- | --- | --- | --- | --- | --- |
| | | Gaussian | Speckle | S & P | Uniform |
| TexQ (Chen et al., 2023) | 50.28 | 33.01 (-34.35%) | 29.80 (-40.74%) | 40.82 (-18.81%) | 39.85 (-20.74%) |
| **SYNQ (Proposed)** | **52.02** | **43.29 (-16.78%)** | **35.62 (-31.53%)** | **46.81 (-10.01%)** | **45.11 (-13.28%)** |

### C.8 ANALYSIS ON THE ROBUSTNESS TOWARDS NOISE

We validate the robustness of SYNQ towards different types of noise. Table 9 compares how TexQ and SYNQ perform in quantization when four distinct noise types are introduced into their synthetic datasets. Specifically, we report the 3bit ZSQ accuracy of a ResNet-18 model pre-trained on ImageNet dataset with four types of noise: Gaussian, speckle, Salt-and-Pepper (S & P), and uniform (Bovik, 2010). We have two observations from the result. First, the low-pass filter effectively minimizes accuracy degradation across various noise types, surpassing the baseline in capacity. Second, the effect of low-pass filter and the influence of noise both vary significantly depending on the noise type. Thus, our future work involves tailoring the noise filtering approach to better handle specific noise types in synthetic datasets.

### C.9 PERFORMANCE REGARDING THE SIZE OF SYNTHETIC DATASET

We analyze the performance variation of SYNQ according to the size of synthetic dataset. Figure 11 shows the 3bit ZSQ accuracy of ResNet-18 model pre-trained on ImageNet dataset. We report the performance while doubling the size of synthetic dataset used for training from 80 to 5,120 images. Note that we compare the model performance trained with 5,120 samples for the main experiments (see Appendix D). We have two observations from the result. First, SYNQ achieves higher performance when trained with a greater number of images. Although the incremental gains begin to drop near 1,000 images and onward, we expect that generating more than 5,120 synthetic images could achieve superior accuracy than the results reported in Table 1. Second, SYNQ outperforms TexQ even when training with only half the dataset, demonstrating the effectiveness of synthesis-aware fine-tuning introduced by SYNQ. In overall, SYNQ shows better performance compared to the baselines, with performance improving as the synthetic dataset size increases.

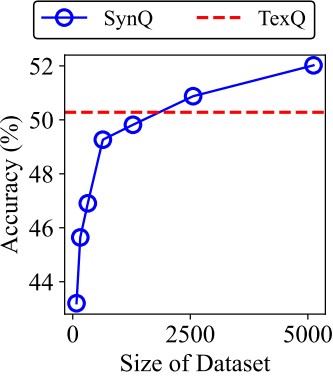

Figure 11: ZSQ accuracy regarding the size of synthetic dataset. See Appendix C.9 for details.

### C.10 VISUALIZATION OF SYNTHETIC DATASET

Although SYNQ is applicable to any ZSQ methods that generate a synthetic dataset, investigating 1) the training set utilized for the highest performance and 2) the effect of the low-pass filter (Idea 1) is essential in understanding SYNQ. Figure 12 presents a visualization of images from three synthetic datasets before and after the low-pass filter. These datasets are generated following the baseline method which is detailed in Appendix D, by three different models: a ResNet-20 model pre-trained on CIFAR-10 dataset (Figures 12a and 12b), a ResNet-20 model pre-trained on CIFAR-100 dataset (Figures 12c and 12d), and a ResNet-18 model pre-trained on ImageNet dataset (Figures 12e and 12f). In order to effectively visualize the effect of the low-pass filter, we set the filtering hyperparameter $D_0$ to 8, 8, and 40 for CIFAR-10, CIFAR-100, and ImageNet datasets, respectively. We have two observations from Figure 12. First, the visualized images show distinct patterns and differences across various classes. Second, the low-pass filter removes noise effectively while preserving essential features in the generated images, which are noticeable especially in lower resolution samples.

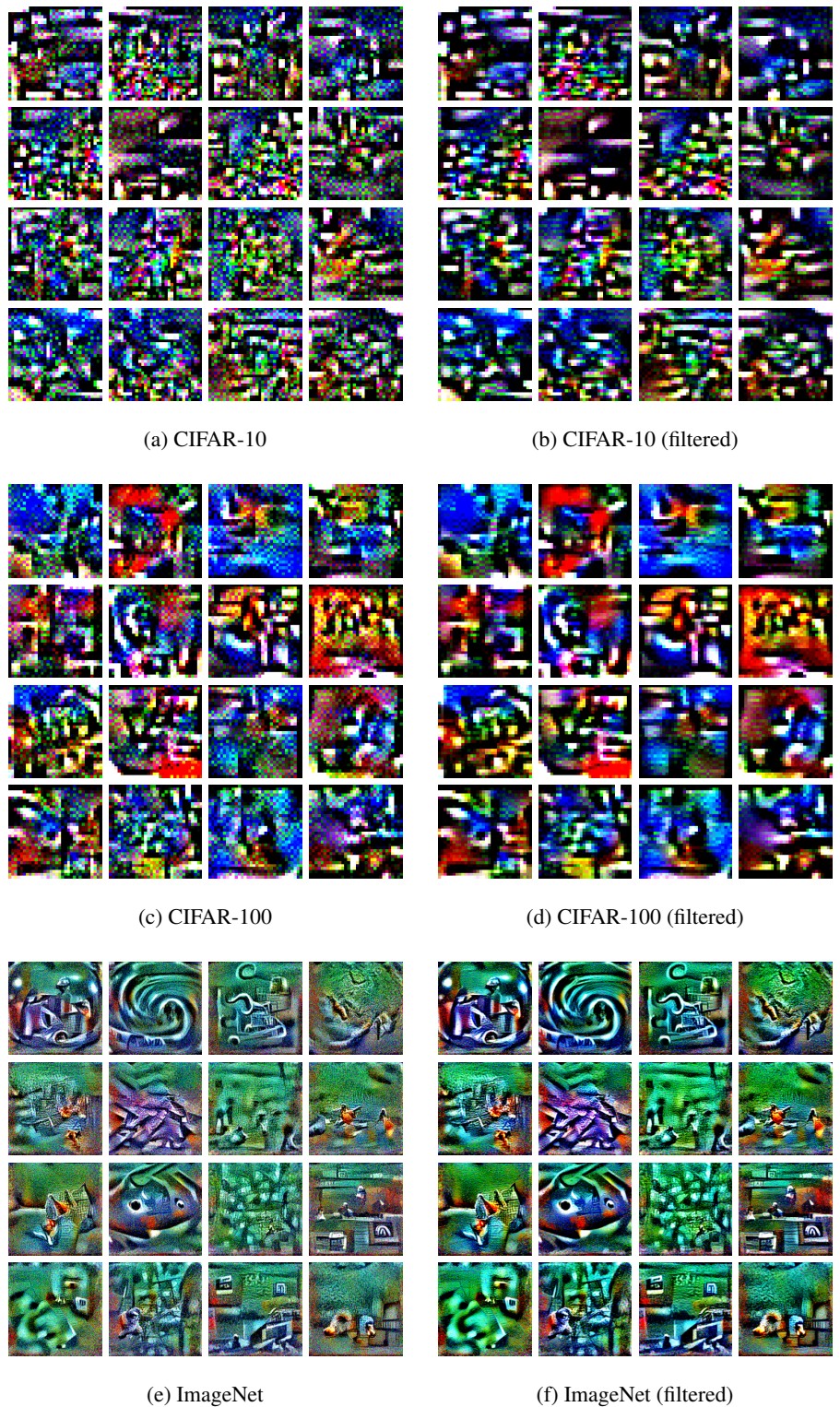

(a) CIFAR-10          (b) CIFAR-10 (filtered)

(c) CIFAR-100          (d) CIFAR-100 (filtered)

(e) ImageNet          (f) ImageNet (filtered)

Figure 12: Visualization of samples within the synthetic dataset before (left) and after (right) the low-pass filter, generated by (a, b) a ResNet-20 model pre-trained on CIFAR-10 dataset, (c, d) a ResNet-20 model pre-trained on CIFAR-100 dataset, and (e, f) a ResNet-18 model pre-trained on ImageNet dataset. See Appendix C.10 for details.

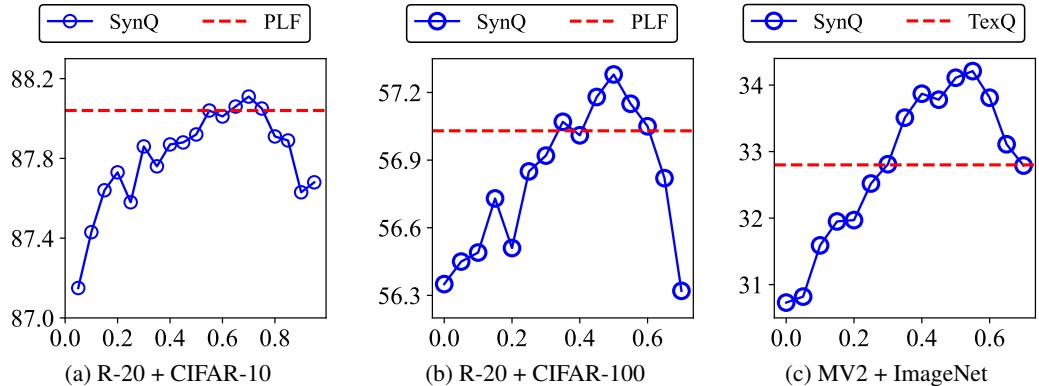

Figure 13: Hyperparameter analysis of difficulty threshold $\tau$ with ResNet-20 (R-20) model pre-trained on CIFAR-10 and CIFAR-100 datasets, and MobileNetV2 (MV2) model pre-trained on ImageNet dataset. See Appendix C.11 for details.

### C.11 FURTHER ANALYSIS ON THE DIFFICULTY THRESHOLD $\tau$

In Section 5.6 and Figure 7, we conduct a hyperparameter analysis to analyze the robustness of SYNQ towards newly introduced hyperparameters. We investigate this aspect across various models and datasets to ensure SYNQ reflects similar tendencies across multiple settings. Figure 13 shows the ZSQ accuracy of (a) a ResNet-20 (R-20) model pre-trained on CIFAR-10 dataset, (b) a ResNet-20 (R-20) model pre-trained on CIFAR-100 dataset, and (c) a MobileNet-V2 (MV2) model pre-trained on ImageNet dataset, with different $\tau$ values. Note that Figure 7(b) introduces the result of a ResNet-18 model on ImageNet dataset. We also depict the performance of the state-of-the-art competitor as a red line for all figures. Note that SYNQ shows similar tendency across different settings, while (a) R-20 + CIFAR-10 maximizes with the $\tau$ value of 0.7. This is because the error rate of pre-trained models (see Figure 3) begins to increase at a higher difficulty level of approximately 0.65 for CIFAR-10, compared to 0.5 for the others. In summary, the optimal $\tau$ should provide a nice trade-off between containing sufficient samples and not using wrong samples.

## D DETAILS ON THE EXPERIMENTAL SETUP

We describe the details on the experimental setup, including datasets, competitors, hyperparameters, implementation, and training.

**Datasets.** We utilize three benchmark datasets, CIFAR-10, CIFAR-100 (Krizhevsky et al., 2009), and ImageNet (ILSVRC 2012) (Deng et al., 2009) to evaluate the classification accuracy of the quantized model obtained by SYNQ. We directly use both CIFAR-10 and CIFAR-100 datasets in TorchVision package. Note that we utilize real datasets only for evaluation purposes.

**Competitors.** We briefly summarize the details of the competitors of SYNQ as follows:

- **GDFQ** (Xu et al., 2020) is the first method to utilize a knowledge-matching generator to produce synthetic data which is guided by both batch normalization statistics loss and cross-entropy loss.

- **ARC** (Zhu et al., 2021) or AutoReCon is a neural architecture search-based image reconstruction method.

- **Qimera** (Choi et al., 2021) uses superposed latent embeddings to generate synthetic boundary supporting samples.

- **AIT** (Choi et al., 2022) improves the loss function and gradients for ARC to generate better samples, which we denote it as AIT + ARC.

- **IntraQ** (Zhong et al., 2022b) highlights the intra-class heterogeneity and retains this property in the synthetic dataset for better performance.

- **AdaSG** (Qian et al., 2023b) plots the ZSQ problem as a zero-sum game between two players, the generator and the quantized network to generate adaptive samples for the synthetic dataset.

- **AdaDFQ** (Qian et al., 2023a) further generalizes AdaSG to adaptively regulate the adaptability of the synthetic samples.

- **HAST** (Li et al., 2023a) pays more attention to difficult samples by generating difficult samples and further promoting the sample difficulty when training the quantized model.

- **TexQ** (Chen et al., 2023) retains the texture feature distributions within the synthetic dataset by using synthetic calibration centers to calibrate samples.

- **PLF** (Fan et al., 2024) evaluates synthetic data to assign pseudo-labels with different reliability to avoid misleading training.

Additionally, we compare with PSAQ-ViT (Li et al., 2022) and Genie (Jeon et al., 2023b) for the ViT (see Section 5.3) and PTQ (see Appendix C.7) experiments, respectively.

**Baseline.** We introduce the baseline method to produce the synthetic dataset for the main results and observations (e.g. Tables 3 and 6). We adopt calibration center synthesis (Chen et al., 2023), difficult sample generation, and sample difficulty promotion (Li et al., 2023a). Producing the synthetic dataset consists of three stages. First, we produce calibration centers following Chen et al. (2023), one center each for all possible classes. Second, we produce the synthetic dataset with two additional losses, hard-sample-enhanced inception loss $\mathcal{L}_{HIL}$ from HAST and layered batch normalization statistics alignment loss $\mathcal{L}_{L-BNS}^{G}$, added on top of Equation 1. Lastly, we attach a perturbation to each image following sample difficulty promotion from HAST, to make generated samples more difficult for the quantized model. We select this baseline that combines only the synthetic dataset production part of the two papers HAST and TexQ, in order to intentionally set baselines only for the first step and replace the existing fine-tuning process with the proposed synthesis-aware fine-tuning. Refer to the original papers for further details.

**Hyperparameters.** We conduct a grid search to validate hyperparameters, and select the set with the best performance. Table 10 reports the searched hyperparameter ranges of SYNQ for the ImageNet dataset experiment. For competitors, we search within the range described in each paper. We conduct 5 iteration for each experiments and report the mean and standard deviation of the results.

Table 10: Hyperparameter ranges for SYNQ.

| Hyperparameter | Range |
| --- | --- |
| $\alpha_1$ | [0.2, 0.4, 0.6, 0.8, 1] |
| $\alpha_2$ | [0.01, 0.04, 0.1] |
| $\alpha^C$ | [0.4, 1, 2.5] |
| $\lambda_P$ | [0.25, 0.5, 0.75, 1, 1.5, 2, 3, 4] |
| $\lambda_{CE}$ | [5, 5e-1, 5e-2, 5e-3] |
| $D_0$ | [20, 40, 60, 80, 100] |
| $\tau$ | [0.5, 0.55, 0.6, 0.65, 0.7] |
| $\lambda_{CAM}$ | [20, 50, 100, 200, 300, 500, 2000] |
| CAM Technique | [CAM, Grad-CAM, Grad-CAM++] |

**Implementation and Machine.** We implement SYNQ with PyTorch and TorchVision libraries in Python. For the other methods, we reproduce the result using their open-source code if possible and implement them otherwise. All of our experiments were done at a workstation with Intel Xeon Silver 4214 and RTX 3090.

**Training Details.** We first generate the calibration centers with a constant learning rate of 0.05, following TexQ (Chen et al., 2023). Then, we optimize samples using the loss function described in Equation 1 with the Adam optimizer to generate the synthetic dataset. This optimizer has a momentum of 0.9 and an initial learning rate of 0.5. The synthetic images are updated over 1,000 iterations, with the learning rate decaying by a factor of 0.1 whenever the loss does not decrease for

50 consecutive iterations. For all datasets, a batch size of 256 is used, resulting in the generation of a total of 5,120 images. For the fine-tuning of the quantized model, the procedure follows Equation 6, employing SGD with a momentum of 0.9 and a weight decay of 1e-4. The batch size is set to 256 for CIFAR-10/100 and 16 for ImageNet. Initial learning rate is searched within the range of {1e-4, 1e-5, 1e-6} and is decayed by a factor of 0.1 over training epochs $n_{ep} = 100$.

**ViT Quantization Experiment.** We compare the ZSQ precision of PSAQ-ViT (Li et al., 2022) with that of SYNQ applied on it. PSAQ-ViT generates the synthetic dataset based on the patch similarity, substituting the batch normalization statistics in CNN models. The pre-trained DeiT-Tiny, DeiT-Small (Touvron et al., 2021), Swin-Tiny, and Swin-Small (Liu et al., 2021b) models on ImageNet dataset is obtained from timm (Wightman, 2019) library. We follow Li et al. (2022) for the experimental setup, where only 32 images are used from the synthetic dataset.

