# OpenReview forum: "SynQ: Accurate Zero-shot Quantization by Synthesis-aware Fine-tuning"
_ICLR.cc/2025/Conference — ICLR 2025 Poster_

### Official Review · Reviewer_7rcv · 2024-10-26

**Soundness:** 3
**Presentation:** 3
**Contribution:** 3
**Rating:** 8
**Confidence:** 4

**Summary:**

They propose SYNQ that targets to overcome the following limitations of current ZSQs:
1. noise in the synthetic dataset; 2. off-target patterns; 3. misguidance by erroneous hard labels.

**Strengths:**

The manuscript exhibits a coherent structure and is straightforward to navigate. Figures 1 through 3 effectively illustrate the key observations and the rationale behind our approach. Notably, the visualization of the Magnitude spectrum in Figure 1 is particularly engaging. To the best of my knowledge, this method is the first zsq to complete experiments on both cnn and vit, which is appreciated.

**Weaknesses:**

1.line382:The header method of Table 1 is incorrectly written as CIFAR dataset
2. line237: The Low-pass filter (Section 4.2) directly modifies the image's impact on the model without using artificial visual features to judge whether it is good or bad. Does Low-pass filters have advantages over the existing ZSQ?
3. Is Fig.2 different at different bit widths/networks? Is this a general situation in ZSQ?
4. Lack of computational cost analysis comparison with state of the art methods.

**Questions:**

The authors could refer to the Weaknesses.

---

> ### Author Response · Authors · 2024-11-21
> **Rebuttal by Authors**
>
> We sincerely appreciate your insightful and constructive review.
> We have carefully taken your insights into account and addressed each point.
> Please kindly refer to the revised manuscript, where the specific updated parts are discussed in detail within the following answers.
> We marked added or modified parts as blue for reviewers’ convenience.
>
> **[W1] Header of Table 1.**
>
> **[A1]**
> We thank the reviewer for carefully checking the manuscript and finding the error in the header of Table 1.
> We modified the first row to identify both the model and dataset of each experiment, and clarified the setup.
>
> **[W2] Advantage of low-pass filters.**
>
> **[A2]**
> As noted by the reviewer, a low-pass filter directly modifies the synthetic image, without any consideration of visual features.
> We exploit this low-pass filter to reduce the noise in the synthetic dataset on the fly by applying traditional techniques.
> The success of this technique clearly shows that mitigating the noise in the synthetic dataset is a key issue that should be addressed in the ZSQ domain.
> We appreciate the reviewer’s suggestion to incorporate visual features and to explore adaptive filtering, which will be part of our future work.
>
> **[W3] Observation (Figure 2) under limited conditions.**
>
> **[A3]**
> We further analyze the CAM pattern discrepancies across different settings in Appendix C.4.
> Table 5 demonstrates that 1) the saliency map varies notably for the quantized models trained on synthetic datasets, and 2) SynQ effectively addresses this issue through CAM alignment (Idea 2).
>
> **[W4] Computational cost comparison with the SOTA methods.**
>
> **[A4]**
> To analyze the impact of the additional computational overhead introduced by SynQ, we added a runtime analysis in Appendix C.2.
> From Figure 8, we observe that the runtime overhead caused by SynQ is minimal, contributing just 82.19% to the overall fine-tuning time on average.
> Overall, SynQ achieves a significant accuracy improvement with only a slight increase in quantization time.

---

> > ### Comment · Reviewer_7rcv · 2024-11-28
> > **Follow up on the low-pass filter**
> >
> > Thank you for your rebuttal to my concerns. I still have an open question. Does the introduction of low-pass filters lead to drastic changes in the labels of the synthetic images? In my previous observations, simple data enhancement (such as flipping) on a synthetic image can drastically reduce the confidence of the synthetic image and even trigger label changes. If a drastic change in the label is caused, will it present a new challenge to the fine-tuning process?

---

> > > ### Author Response · Authors · 2024-11-28
> > >
> > > Dear Reviewer 7rcv,
> > >
> > > We appreciate your insightful question regarding the drastic label changes induced by image augmentations and the introduction of the low-pass filter.
> > >
> > > To investigate the effect of image augmentation, we evaluate the classification accuracy of both full-precision (FP) and 4bit (W4A4) quantized models on the ImageNet dataset.
> > > Next, we apply 15 augmentations from RandAugment [1] and report the accuracy drop as follows:
> > >
> > > |Aug. Type | Augmentation | ViT-B (FP) | ViT-B (W4A4) | DeiT-B (FP) | DeiT-B (W4A4) |
> > > |:-----:|:-----:|:-----:|:-----:|:-----:|:-----:|
> > > | - | Original | 84.536 | 72.288 | 81.796 | 75.812 |
> > > | Vertical | ResizedCrop | 78.036 (-7.69%) | 62.664 (-13.31%) | 74.934 (-8.39%) | 66.876 (-11.79%) |
> > > | | Yflip | 77.352 (-8.50%) | 58.778 (-18.69%) | 72.934 (-10.83%) | 63.368 (-16.41%) |
> > > | | Rotate | 75.362 (-10.85%) | 48.318 (-33.16%) | 65.900 (-19.43%) | 51.858 (-31.60%) |
> > > | |Affine | 75.454 (-10.74%) | 48.278 (-33.21%) | 66.112 (-19.17%) | 51.420 (-32.17%) |
> > > | Color | Perspective | 82.620 (-2.27%) | 67.960 (-5.99%) | 80.602 (-1.46%) | 71.776 (-5.32%)|
> > > | | Solarize | 80.636 (-4.61%) | 64.196 (-11.19%) | 79.688 (-2.58%) | 71.764 (-5.34%) |
> > > | | Grayscale | 81.182 (-3.97%) | 61.900 (-14.37%) | 78.178 (-4.42%) | 69.580 (-8.22%) |
> > > | | Hue | 77.698 (-8.09%) | 58.192 (-19.50%) | 72.488 (-11.38%) | 61.964 (-18.27%) |
> > > | Others | Xflip | 84.344 (-0.23%) | 72.028 (-0.36%) | 81.754 (-0.05%) | 75.650 (-0.21%) |
> > > | | Posterize | 84.322 (-0.25%) | 72.148 (-0.19%) | 81.726 (-0.09%) | 75.558 (-0.34%) |
> > > | | Saturation | 84.160 (-0.44%) | 71.486 (-1.11%) | 81.652 (-0.18%) | 75.292 (-0.69%) |
> > > | | Contrast | 84.110 (-0.50%) | 71.122 (-1.61%) | 81.538 (-0.32%) | 74.956 (-1.13%) |
> > > | | Brightness | 84.018 (-0.61%) | 70.774 (-2.09%) | 81.544 (-0.31%) | 74.836 (-1.29%) |
> > > | | Invert | 84.028 (-0.60%) | 68.674 (-5.00%) | 81.502 (-0.36%) | 74.588 (-1.61%) |
> > > | (Average) | RandAugment | 83.278 (-1.49%) | 69.898 (-3.31%) | 80.876 (-1.12%) | 73.510 (-3.04%) |
> > >
> > > From these results, we derive two key observations.
> > > Firstly, augmentations involving vertical or color transformations cause a significant accuracy drop of 10-30% in quantized models.
> > > Notably, the models exhibit robustness to horizontal (x-axis) perturbations but are sensitive to vertical (y-axis) changes.
> > > Secondly, pre-trained models in full precision also exhibit performance degradation under these augmentations.
> > > This suggests that pre-trained models may not have been sufficiently trained to be robust against such changes, rendering them particularly vulnerable to quantization.
> > > In summary, certain augmentations involving vertical or color transformations degrade the classification performance of pre-trained models and, more so, quantized models, ultimately lowering the confidence in synthetic images as noted by the reviewer.
> > >
> > > Consistent with our observations, applying a low-pass filter does not substantially influence class label confidence, given its lack of vertical or color-based changes.
> > > However, excessive filtering may degrade image quality, impacting performance; thus, we carefully select the optimal filtering hyperparameter $D_0$​ to achieve the best results.
> > >
> > > That said, following the reviewer’s insight, the performance degradation of quantized models from various image augmentations seems to be a key issue worth further exploration.
> > > We are grateful for the reviewer’s comment and will incorporate this insight in our future studies.
> > >
> > > With best gratitude,
> > >
> > > Authors of Submission 6292
> > >
> > > **References.**
> > >
> > > [1] Cubuk et al., “RandAugment: Practical Automated Data Augmentation with a Reduced Search Space”, NeurIPS 2020

---

> > > > ### Comment · Reviewer_7rcv · 2024-12-01
> > > >
> > > > Thank you for your comprehensive feedback. I hope the authors to release code to foster community communication.

---

> ### Author Response · Authors · 2024-12-02
>
> Dear Reviewer 7rcv,
>
> We are grateful for your constructive feedback and are delighted to hear that our efforts have resulted in an overall improvement.
> Regarding the official implementation of SynQ, it is available online through both the supplementary materials and via https://anonymous.4open.science/r/SynQ_off, as mentioned in line 107 of the manuscript.
> For the brief experiment discussed above, our implementation relies on the official codes of RepQ-ViT [1] and RandAugment [2].
> Overall, we are deeply thankful for your feedback and are looking forward to further perfecting our research.
>
> With our thanks,
>
> Authors of Submission 6292
>
> **References.**
>
> [1] Li et al., “Repq-vit: Scale reparameterization for post-training quantization of vision transformers”, ICCV 2023
>
> [2] Cubuk et al., “RandAugment: Practical Automated Data Augmentation with a Reduced Search Space”, NeurIPS 2020

---

### Official Review · Reviewer_jAz6 · 2024-11-03

**Soundness:** 3
**Presentation:** 2
**Contribution:** 3
**Rating:** 6
**Confidence:** 5

**Summary:**

This work points out several problems that prior works of Data-free quantization (DFQ) have.
First, synthesized images are noisy compared to their real counterparts.
Second, models quantized with synthesized images tend to predict based on incorrect image patterns.
In addition, the paper claims that using hard labels on hard samples can cause misguidance.

To resolve these problems, the paper proposes three methods.
- The paper revisits classical signal processing and removes the noise of generated images with a low-pass filter.
- To align activation maps between a full precision model and a quantized model, the paper proposes to use grad CAM as a loss function.
- By considering model outputs as the difficulty of the sample, the paper proposes to omit CE loss if the input is a difficult sample.

**Strengths:**

- The paper tackles the limitations of previous works well.
- The paper tries to denoise synthesized images with a loss-pass filter. This idea is a good point that highlights the importance of classical techniques and theories in the recent AI era.
- The paper identifies the off-target prediction problem that occurs only in the data-free quantization scenario. It is a good suggestion that analyzes the performance of a quantized model with grad GAM and uses it as a loss function.
- The paper executes various experiments and ablation studies for validating proposals.

**Weaknesses:**

- The paper refers to the limitations of previous works too much. The same content is repeated over 3 times.
- The investigation and analysis of prior works is insufficient.
    - The paper notes that using hard labels can be harmful to the performance of quantized models, pointing out that previous works used both hard labels (CE loss) and soft labels (KL divergence loss). It can be a novelty that determines the usage of CE loss according to difficulty. However, there already exist several works that use a soft label instead of a hard label. For instance, Qimera proposed to use the coefficient of superposed latent embedding as soft labels. AIT also pointed out the importance of soft labels and used soft labels only for the loss function.
    - The results of current state-of-the-art works are omitted. In the CNN domain, GENIE shows better performance than this work. Also, in transformer variants, PSAQ-ViT V2 shows better results. Those works should be addressed.
- Generated images with SynQ can help understand the superiority of the proposal.  Please attach generated images with SynQ (before and after applying the low-pass filter)

**Questions:**

- The reviewer thinks that off-target prediction problems may occur under the DFQ scenario, not general quantization scenarios. Nevertheless, did the authors examine whether the problem occurred with real data?
- What is the experimental setting of the baseline in Table 3 such as the quantization algorithm? (the same result is not in the Table 1)
- In Figure 7,
    - Does this tendency still hold with other models and datasets? For instance, the distribution of model outputs that are used as a measurement of difficulty can be different if the number of classes differs. With various models (e.g., ResNet, MobileNet, etc) and datasets (e.g., CIFAR10, CIFAR100, ImageNet), is the optimal $\tau$ always 0.5 or values close to it?
    - The magnitude of $\lambda_{CAM}$ in (a) are much larger than those of $\lambda_{CE}$. Is the magnitude of CAM loss on average much smaller than that of $\lambda_{CE}$ loss?
- In Table 5 of the appendix,
    - Those experiments are based on 3 works that use a generator. However, SynQ adopts noise optimization for generating images. Why aren’t other works that adopt noise optimization addressed?
    - Those 3 works are improved with SynQ. However, they are worse than SynQ itself. Can the authors’ opinions about this provided?
    - How about applying SynQ to other works based on noise optimization?

---

> ### Author Response · Authors · 2024-11-21
> **Rebuttal by Authors (1)**
>
> We sincerely appreciate your high-quality review with constructive comments.
> We have carefully taken your insights into account and addressed each point.
> Please kindly refer to the revised manuscript, where the specific updated parts are discussed in detail within the following answers.
> We marked added or modified parts as blue for reviewers’ convenience.
>
> **[W1] Repetition of three major limitations.**
>
> **[A1]**
> We understand the reviewer's point regarding the repeated references to similar content across the paper.
> That said, we consider these challenges to be a key aspect of the paper and feel that reiterating their importance is necessary to convey their central role in this paper.
> We aim to use a top-down structure so that even readers skimming the paper could easily grasp the key messages, recognizing that this naturally results in some repetition, though thorough readers like the responsible reviewer may find it less necessary.
> In this context, we intentionally unified the titles of challenges and their corresponding observations to avoid potential confusion, while providing varying perspectives within the content of each.
> We hope this clarifies our intentions and aligns with the reviewer's valuable observations.
>
> **[W2] Novelty of ‘Soft Labels for Difficult Samples’ (Idea 3).**
>
> **[A2]**
> As the reviewer pointed out, our proposed idea of using ‘Soft labels for difficult samples’ presents the novelty of dynamically adjusting the usage of CE loss depending on sample difficulty.
> To review the previous works, one set of papers such as Qimera [1] and AIT [2], support the use of soft labels rather than hard labels. Another set of papers, such as HAST [3], emphasizes the importance of the difficulty distribution of samples within the synthetic dataset.
> Inspired by the definition of difficulty in Equation 3 and the observation in Figure 3, we combined the insights from both sets of research and proved that the decision to use hard labels based on difficulty is both intuitive and experimentally supported.
> While the aforementioned methods all determine the use of hard and soft labels uniformly across samples, loss splitting based on sample difficulty distinguishes our work from existing approaches and makes a novel contribution.
>
> **[W3(a)] Comparison with Genie [4].**
>
> **[A3]**
> The key distinction between Genie [4] and our proposed SynQ is in the quantization approach; QAT methods including SynQ adopt min-max quantization, while Genie adapts advanced techniques such as LSQ [5], QDrop [6], and BRECQ [7].
> We provide a detailed comparison on the settings of Genie and SynQ to explain why Genie is neglected from our competitors in the main experiments (Table 1) in Appendix C.7.
> Due to the differences in quantization strategies and experimental conditions, evaluating Genie alongside zero-shot QAT methods is challenging.
> Then, we integrate SynQ with Genie and evaluate its evaluation performance to compare the performance with Genie and highlight the broad applicability of our proposed method.
> The results in Table 8 demonstrate the superiority of SynQ, showcasing its compatibility with diverse quantization techniques such as zero-shot PTQ.
>
> **[W3(b)] Additional experiments on ViT baselines.**
>
> **[A4]**
> We agree on the necessity of experimenting with SOTA models in paper writing.
> Regarding the Zero-Shot Quantization (ZSQ) of Vision Transformers (ViTs), three papers have been published to date: PSAQ-ViT [8], PSAQ-ViT V2 [9], and CLAMP-ViT [10].
> However, we have only been able to conduct experiments by applying SynQ to PSAQ-ViT, as outlined in Section 5.3.
> Despite our intention to experiment with SynQ on both PSAQ-ViT V2 and CLAMP-ViT, we encountered two issues that prevented us from doing so: 1) neither paper has an official code implementation available in their public repositories, and 2) we were unable to reproduce the results reported in those papers.
> For PSAQ-ViT V2, while it was claimed to be released along with the PSAQ-ViT code, we found only the PSAQ-ViT code when we checked the link (https://github.com/zkkli/PSAQ-ViT), and observed that there have been no updates since the last commit two years ago.
> For CLAMP-ViT, the official code repository (https://github.com/georgia-tech-synergy-lab/CLAMP-ViT) includes only the mixed-precision model weights and evaluation code, with no code for the synthetic dataset generation or quantization process.
> The failure to reproduce the experiments is due to the lack of detailed information regarding certain hyperparameters and other experimental settings, which made it impossible to replicate the reported results.
> Given that SynQ is a synthesis-aware fine-tuning technique, it is applicable to all methods that generate synthetic datasets.
> We further aim to continue following relevant studies on ZSQ of ViTs and explore the impact of SynQ on other methods, beyond PSAQ-ViT.

---

> ### Author Response · Authors · 2024-11-21
> **Rebuttal by Authors (2)**
>
> **[W4] Illustration of the synthetic dataset.**
>
> **[A5]**
> We appreciate the reviewer’s gentle recommendation on the visualization of synthesized data to help understand the superiority of SynQ.
> In Appendix C.10, we compare the images from three synthetic datasets before and after the low-pass filter.
> From Figure 12, we have two observations.
> First, the visualized images show distinct patterns and differences across various classes.
> Second, the low-pass filter removes noise effectively while preserving essential features in the generated images, especially noticeable in lower resolution samples.
>
> **[Q1] Investigating “off-target prediction” when training with real images.**
>
> **[A6]**
> Although intuitive and persuasive, Figure 2 has a limitation that we observe under limited conditions.
> In Appendix C.4, we extend this to entire synthetic datasets generated from various methods by evaluating the CAM discrepancy, to validate that the challenge of “off-target prediction” is a common issue.
> Also, we investigate quantized models that are fine-tuned with real datasets to check if the challenge is induced by the usage of synthetic datasets.
> Table 5 highlights that training with synthetic datasets exacerbates CAM discrepancy,
> as the reviewer pointed out.
>
> **[Q2] Baseline of SynQ.**
>
> **[A7]**
> As discussed in Section 4.5, we adopt calibration center synthesis [11], difficult sample generation [3], and sample difficulty promotion [3] as baseline to produce the synthetic dataset for the best performance.
> This baseline method is utilized for the main results and observations including Tables 1, 3, 5, and Figure 7.
> To clarify the experimental settings and for better reproducibility, we provide the step-by-step implementation details of the baseline in Appendix D.
> As we combine only the synthetic dataset generation part of two papers HAST and TexQ for baseline, the performance of this baseline is not listed in Table 1.
>
> **[Q3(a)] Further analysis on the difficulty threshold $\tau$.**
>
> **[A8]**
> We conduct a hyperparameter analysis on difficulty threshold $\tau$ under different settings in Appendix C.11.
> Figure 13 depicts the ZSQ accuracy with different $\tau$ values of (a) a ResNet-20 model pre-trained on CIFAR-10 dataset, (b) a ResNet-20 model pre-trained on CIFAR-100 dataset, and (c) a MobileNet-V2 model pre-trained on ImageNet dataset.
> As shown in the figure, SynQ shows similar tendency across different settings, while (a) maximizes with the $\tau$ value of 0.7.
> This is because the error rate of pre-trained models in Figure 3 begins to increase at a higher difficulty level of approximately 0.65 for CIFAR-10, compared to 0.5 for the others.
> In summary, SynQ shows a common trend of its performance regarding $\tau$ across various settings, where the optimal $\tau$ should provide a nice trade-off between containing sufficient samples and not using wrong samples.
>
> **[Q3(b)] Magnitude comparison between CAM and cross-entropy losses.**
>
> **[A9]**
> We compare the magnitude of CE and CAM losses in the table below:
>
> | Dataset  | $\mathcal{L}_{CE}$ | $\mathcal{L}_{CAM}$ | Ratio |
> |:------------|:--------:|:--------:|:----------:|
> | CIFAR-10 | 18.857 | 0.014 | 1337.62 |
> | CIFAR-100 | 43.733 | 0.042 | 1047.80 |
> | ImageNet | 51.026 | 0.039 | 1297.61 |
>
> We report the 3bit quantization result of a ResNet-20 model pre-trained on the CIFAR-10 dataset, a ResNet-20 model pre-trained on the CIFAR-100 dataset, and a ResNet-18 model pre-trained on the ImageNet dataset.
> The ratio of CE loss to CAM loss is around 1K for all three cases.
> Thus, for the best-performing case, $\lambda_{CAM}$ is set much larger than $\lambda_{CE}$ to balance the scales of the two terms during computation.

---

> ### Author Response · Authors · 2024-11-21
> **Rebuttal by Authors (3)**
>
> **[Q4] Application on different baselines.**
>
> **[A10]**
> We appreciate the suggestion of the reviewer to add extra experimental results applying SynQ on noise optimization baselines.
> In Appendix C.6, we evaluate the 3bit ZSQ accuracy of the ResNet-18 model pre-trained on the ImageNet dataset, comparing the performance with and without SynQ (see Table 7).
> Beyond the existing results on three generator-based baselines (GDFQ [12], Qimera [1], and AdaDFQ [13]), we add extra results on three noise optimization baselines: IntraQ [14], HAST [3], and TexQ [11].
> SynQ shows consistent performance enhancement through all baselines and bitwidth, with up to 8.11%p for noise optimization baselines.
>
> Additionally, the reviewer asked the authors’ opinion on the performance of generator-based models.
> Specifically, when SynQ is applied to these methods, their performance is lower than that reported for SynQ in Table 1.
> The lower performance observed would be attributed to the synthetic dataset's quality; SynQ yields better results when fine-tuned with a higher-quality dataset.
> The accuracy of the generator-based methods is lower than that of other noise optimization methods, even after applying SynQ.
> This suggests that the image quality of these methods is inferior to the baseline.
>
> **References.**
>
> [1] Choi et al., “Qimera: Data-free Quantization with Synthetic Boundary Supporting Samples”, NeurIPS 2021
>
> [2] Choi et al., “It's All In the Teacher: Zero-Shot Quantization Brought Closer to the Teacher”, CVPR 2022
>
> [3] Li et al., “Hard Sample Matters a Lot in Zero-Shot Quantization”, CVPR 2023
>
> [4] Jeon et al., “Genie: show me the data for quantization”, CVPR 2023
>
> [5] Esser et al., “Learned Step Size Quantization”, ICLR 2020
>
> [6] Wei et al., “QDrop: Randomly Dropping Quantization for Extremely Low-bit Post-Training Quantization”, ICLR 2022
>
> [7] Li et al., “BRECQ: Pushing the Limit of Post-Training Quantization by Block Reconstruction”, ICLR 2021
>
> [8] Li et al., “Patch Similarity Aware Data-Free Quantization for Vision Transformers”, ECCV 2022
>
> [9] Li et al., “PSAQ-ViT V2: Towards Accurate and General Data-Free Quantization for Vision Transformers”, TNNLS 2023
>
> [10] Ramachandran et al., “CLAMP-ViT: Contrastive Data-Free Learning for Adaptive Post-Training Quantization of ViTs”, ECCV 2024
>
> [11] Chen et al., “TexQ: Zero-shot Network Quantization with Texture Feature Distribution Calibration”, NeurIPS 2023
>
> [12] Xu et al., “Generative Low-bitwidth Data Free Quantization”, ECCV 2020
>
> [13] Qian et al., “Adaptive Data-Free Quantization”, CVPR 2023
>
> [14] Zhong et al., “IntraQ: Learning Synthetic Images with Intra-Class Heterogeneity for Zero-Shot Network Quantization”, CVPR 2022

---

> ### Comment · Reviewer_jAz6 · 2024-11-22
>
> **[A1 Comment]** :  Thanks to the authors’ response, the reviewer is able to understand why the authors repeated those contents. However, the reviewer just thought that if the authors had only summarized the limitations, rather than explaining them in detail repeatedly, readers who skim the paper can go back and reread them and the additional space can be used with other experiments and analyses.
>
> **[A2 Comment]** The authors showed well that using soft labels with hard samples can enhance performance of a quantized model. However, it will be helpful to describe where the idea got inspiration from and what has changed like below.
>
> - Several previous work adopted soft labels, but applying soft labels to hard samples only performs better.
> - Model outputs can be good proxies for measuring sample difficulty, with the proposed soft label loss.
>
> **[A3 / A4 Comment]** The reviewer understands the difficulty that the authors mentioned. Thank the authors for additional experiments to validate the proposal.
>
> **[A5 Comment]** The contour of generated images in SynQ is similar to that in other works, while it is conspicuous that the noise is removed with low-pass filter. This can be a ground of Table 3. Thank the authors for the visualization.
>
> **[A6 / A7 Comment]** Thank the authors for clearly addressing the questions through the experiment and providing answers.
>
> **[A8 Comment]** As the reviewer expected, optimal points of $\tau$ differ according to model and dataset. Nevertheless, it is a good point that the overall tendency of performance change caused by fine-tuning $\tau$ is consistent. The reviewer appreciates the authors demonstrating	it through experiments.
>
> **[A9 Comment]** The reviewer presumed that the magnitude of CAM loss is small because $\lambda_{CAM}$ is very large, and the authors show that with an experiment. Even though $\lambda_{CAM}$ is very small, reducing it helps improving performance. It seems to be a good point to address in the future that those results can be achieved only with optimizing CE loss and CAM loss simultaneously, or CE loss and CAM loss are orthogonal. (CAM loss is critical for generalization ability of a quantized model) Thank the authors for providing such valuable insight.
>
> With the authors' sincere answers, the reviewer has decided to raise the score to 6.

---

> > ### Author Response · Authors · 2024-11-23
> >
> > Dear Reviewer jAz6,
> >
> > We deeply appreciate your valuable feedback and are happy that our revisions have improved the overall quality.
> > Please feel free to provide additional feedback or ask any questions as needed.
> > We are truly grateful for your comments and are eager to continue improving and refining our research.
> >
> > With our thanks,
> >
> > Authors of Submission 6292

---

### Official Review · Reviewer_bwpy · 2024-11-03

**Soundness:** 3
**Presentation:** 3
**Contribution:** 3
**Rating:** 6
**Confidence:** 4

**Summary:**

The paper proposed a synthesis-aware fine-tuning method, SYNQ, to improve zero-shot quantization (ZSQ) performance. SYNQ defines the issues of ZSQ as follows: 1) high-frequency noise in the generated synthetic dataset, 2) predictions based on off-target patterns, and 3) misguidance by hard labels. SYNQ effectively addresses these issues to improve ZSQ performance through the use of a low-pass filter, CAM alignment, and hard label filtering.

**Strengths:**

1. The observations regarding the three limitations of ZSQ are interesting, and the proposed method appears feasible.
2. The performance is validated through a variety of experiments. Specifically, experiments were conducted to verify the performance of SYNQ by comparing it with various ZDQ baselines on not only CNN-based models but also ViT-based models.
3. The detailed analyses of the three components of SYNQ enhance the persuasiveness of the methodology.
4. This paper is well-written and easy to follow.

**Weaknesses:**

1. Although the observations presented in the paper are interesting, most of the experimental evidence provided was gathered under limited conditions. For instance, in Figure 5, experiments were shown only for TexQ among various baseline models, and the analysis for CIFAR-10 and CIFAR-100 used as benchmarks in Table 1 was omitted.
2. In Figure 2, the heat map is shown only one sample image.

For these reasons, it is difficult to be certain whether the presented observations are phenomena that can be observed only in limited baselines and datasets or are generally seen across ZSQ methods. Therefore, the authors should provide experimental evidence across various baselines and datasets beyond the limited settings.

**Questions:**

1. While SYNQ has been evaluated on W3 and W4, how does it perform under extremely low-bit (e.g., 2-bit) conditions? For example, GENIE [1], one of the ZSQ methods, demonstrated performance not only on W3 and W4 but also on W2. It would be beneficial to add it as a baseline and show performance in low-bit settings as well.
2. What is the performance variation according to the size of the generated synthetic dataset?

[1] Jeon et al., "GENIE: Show Me the Data for Quantization. ", CVPR 2023.

---

> ### Author Response · Authors · 2024-11-21
> **Rebuttal by Authors**
>
> We sincerely appreciate your high-quality review with constructive comments.
> We have carefully taken your insights into account and addressed each point.
> Please kindly refer to the revised manuscript, where the specific updated parts are discussed in detail within the following answers.
> We marked added or modified parts as blue for reviewers’ convenience.
>
> **[W1/W2] Observations under limited conditions**
>
> **[A1]**
> We understand the reviewer's concern that our observations might be applicable only under certain conditions, raising doubts about whether the phenomena are truly widespread in the domain.
> To address this concern, we present additional experiments across a range of baseline methods, models, and datasets in Appendices C.3 and C.4, showing that the phenomena are consistently observed.
>
> First, we show that noise in the synthetic dataset remains a widespread issue in ZSQ by extending the Figure 5 experiment across different settings (Appendix C.3).
> Figures 9 and 10 highlight the pervasive nature of noise in synthetic datasets and the consistent impact of low-pass filtering in numerous settings.
>
> Second, we analyze the CAM pattern discrepancies across different baselines trained with synthetic and real datasets (Appendix C.4).
> Table 5 demonstrates that 1) the saliency map varies notably for the quantized models trained on synthetic datasets, and 2) SynQ effectively addresses this issue through CAM alignment (Idea 2).
>
> These results demonstrate that the challenges we identified are critical issues that should be explored crucially in the zero-shot quantization domain.
>
> **[Q1] Evaluation on low-bit conditions.**
>
> **[A2]**
> We investigate the performance of SynQ in low-bit conditions in Appendix C.7.
> While following the experimental setting of Genie [1], we compare the ZSQ accuracy of Genie with and without SynQ in Table 8.
> SynQ successfully enhances the performance of zero-shot PTQ methods, with higher improvements in lower-bit conditions such as W2A2 or W4A4.
>
> **[Q2] Performance regarding the size of synthetic dataset.**
>
> **[A3]**
> We appreciate the reviewer’s gentle recommendation to investigate the performance variation according to the size of the generated synthetic dataset.
> We analyze this in Appendix C.9, where Figure 11 shows the 3bit ZSQ accuracy of the ResNet-18 model pre-trained on ImageNet dataset.
> SynQ demonstrates steadily increasing ZSQ accuracy with larger training datasets, outperforming TexQ even with only half the images.
> This result underscores the scalability and efficiency of SynQ to achieve high accuracy even with constrained training data.
>
> **References.**
>
> [1] Jeon et al., “Genie: show me the data for quantization”, CVPR 2023

---

> > ### Comment · Reviewer_bwpy · 2024-11-25
> > **Response to Author's Rebuttal**
> >
> > Thank you for your rebuttal to my concerns. Most of my concerns are addressed, and I raised my score.

---

> > > ### Author Response · Authors · 2024-11-25
> > >
> > > Dear Reviewer bwpy,
> > >
> > > We greatly value your thoughtful comments and are pleased to hear that our updates have addressed your concerns effectively.
> > > We welcome any additional thoughts or inquiries you might have at any time.
> > > Thank you again for your feedback; your valuable feedback inspires us to keep working on improving and perfecting our research.
> > >
> > > Sincerely,
> > >
> > > Authors of Submission 6292

---

### Official Review · Reviewer_L8UP · 2024-11-07

**Soundness:** 3
**Presentation:** 3
**Contribution:** 3
**Rating:** 6
**Confidence:** 4

**Summary:**

The paper presents SYNQ (Synthesis-aware Fine-tuning for Zero-shot Quantization), a novel framework designed to address the challenges associated with zero-shot quantization (ZSQ) of pre-trained models, particularly in scenarios where training data is inaccessible due to privacy or security concerns. SYNQ tackles three main issues: noise in synthetic datasets, off-target pattern predictions, and misguidance from erroneous hard labels. The proposed method employs a low-pass filter to reduce noise, optimizes class activation map (CAM) alignment to ensure correct image region prediction, and uses soft labels for difficult samples to prevent misguidance. The authors show that SYNQ achieves state-of-the-art accuracy in image classification tasks compared to existing ZSQ methods.

**Strengths:**

1. SYNQ offers a unique solution to the problem of quantizing models without access to training data, which is a significant contribution to deploying neural networks on edge devices.
2. Addressing Key Challenges: The paper clearly identifies and addresses three major challenges in ZSQ, providing a comprehensive approach to improving the accuracy of quantized models.
3. Empirical Validation: Extensive experiments demonstrate SYNQ's effectiveness, showing improvements in classification accuracy over existing methods.

**Weaknesses:**

1. While the paper focuses on image classification, it's unclear how SYNQ would perform in other tasks such as object detection or segmentation.
2. The paper could provide more details on the computational overhead introduced by SYNQ, especially the impact of the low-pass filter and CAM alignment.
3. The paper could benefit from a deeper analysis of SYNQ's robustness to different types and levels of noise in synthetic datasets.

**Questions:**

1. How does SYNQ handle different types of noise, and is its performance consistent across various noise levels? Before and after the low-pass filter, what is the changes of generated images?
2. There are more related papers should be included, such as 'Data-Free Learning of Student Networks', ‘Data-free network quantization with adversarial knowledge distillation’ and others.

---

> ### Author Response · Authors · 2024-11-21
> **Rebuttal by Authors (1)**
>
> We sincerely appreciate your insightful and constructive review.
> We have carefully taken your insights into account and addressed each point.
> Please kindly refer to the revised manuscript, where the specific updated parts are discussed in detail within the following answers.
> We marked added or modified parts as blue for reviewers’ convenience.
>
> **[W1] Application to other tasks.**
>
> **[A1]**
> To begin with, we thank the reviewer for the great idea to extend SynQ to other target tasks, such as object detection or segmentation.
> Applying SynQ to tasks beyond image classification would clearly showcase the wide applicability and broad versatility of our proposed method.
> However, zero-shot quantization of vision models for tasks other than image classification is still an unexplored area, with only one recent work [1] submitted to ICLR 2025 addressing this.
> Unfortunately, we were unable to apply SynQ on the given setting as 1) the work is still under academic peer review, 2) it lacks official code implementation and project page, and 3) the rebuttal phase does not provide sufficient time for our own implementation and validation.
> We will actively pursue further studies on these areas and explore expanding SynQ to other target tasks in our future work.
>
> **[W2] Details on the computational overhead.**
>
> **[A2]**
> In our submitted manuscript, we discuss the computational complexity of SynQ in Theorem 1 and its proof in Appendix C.1.
> As the reviewer pointed out, we agree that our proof could have been more detailed.
> Accordingly, we provide a more detailed analysis of the computational complexity in the updated proof included in Appendix C.1 of the revised manuscript.
> As a result, the computational complexity of the low-pass filter and CAM alignment is $O(NZlogZ)$ and $O(NLT_{\theta})$, respectively, where $N$ is the size of the synthetic dataset, $Z \times Z$ is the input dimension, $L$ is the layer count, and $O(T_{\theta})$ indicates the inference complexity of given pre-trained model.
> Furthermore, to analyze the actual impact of the additional computational overhead introduced by SynQ, we added a runtime analysis in Appendix C.2.
> The runtime overhead caused by SynQ is minimal, contributing just 82.19% to the overall fine-tuning time on average.
> Overall, SynQ achieves a significant accuracy improvement with only a slight increase in quantization time.
>
> **[W3/Q1] Robustness towards different types of noise.**
>
> **[A3]**
> We first clarify our main challenge of “noise in the synthetic dataset” to mitigate possible misunderstandings, then investigate the robustness of SynQ and the impact of the low-pass filter with further experiments.
> The novelty of our first challenge lies in observing that synthetic datasets generated by existing ZSQ methods exhibit greater sharpness and noise levels compared to real datasets (see Figures 1, 5, 9, and 10).
> This inherent noise stems from excessive high-frequency components in the generated samples, disrupting the fine-tuning process.
> We exploit a low-pass filter (Idea 1) on the fly, utilizing traditional methods to effectively mitigate noise.
> The demonstrated success of this approach emphasizes that addressing noise in synthetic datasets is a crucial challenge in the ZSQ domain.
>
> To validate the robustness of SynQ, we evaluate how ZSQ accuracy decreases when different types of noise are intentionally introduced into the synthetic dataset, as detailed in Appendix C.8.
> Table 9 demonstrates that the low-pass filter effectively minimizes accuracy degradation across various noise types, surpassing the baseline in capacity.
>
> To further verify the impact of the low-pass filter, we present additional experiments in Appendices C.3 and C.10.
> Figure 10 illustrates the amplitude distribution of synthetic datasets, comparing the results before and after applying the low-pass filter.
> The results highlight the consistent impact of low-pass filtering in numerous settings.
> Figure 12 provides a visualization of the generated images with different baseline methods and datasets, showing how the low-pass filter behaves at the image level.
> We observe that the low-pass filter removes noise effectively while preserving essential features in the generated images, which are especially noticeable in lower-resolution samples.
> These results highlight the consistent impact of low-pass filtering in numerous settings.
>
> That said, as the reviewer pointed out, the current approach lacks consideration for the specific types of noise in the synthetic dataset.
> We thank the reviewer for the valuable comment; we will reflect on this insight and focus on improving the noise filtering approach in our future work.
>
> **[Q2] Including more related papers.**
>
> **[A4]**
> Following the reviewer’s suggestion, we expanded our related work in Section 6 to include DAFL [2] and DFQ-AKD [3], along with other relevant works such as FDDA [4], QALoRA [5], Mr.BiQ [6], RepQ-ViT [7], Genie [8], QDrop [9], etc.

---

> ### Author Response · Authors · 2024-11-21
> **Rebuttal by Authors (2)**
>
> **References.**
>
> [1] ICLR 2025 Conference Submission4776 Authors, “Zero-shot Quantization for Object Detection”, Submitted to ICLR 2025, https://openreview.net/forum?id=XNr6sexQGj
>
> [2] Chen et al., “Data-free learning of student networks”, CVPR 2019
>
> [3] Choi et al., “Data-free network quantization with adversarial knowledge distillation”, CVPRW 2020
>
> [4] Zhong et al., “Fine-grained data distribution alignment for post-training quantization”, ECCV 2022
>
> [5] Xu et al., “Qa-lora: Quantization-aware low-rank adaptation of large language models”, ICLR 2024
>
> [6] Jeon et al., “Mr. biq: Post-training non-uniform quantization based on minimizing the reconstruction error”, CVPR 2022
>
> [7] Li et al., “Repq-vit: Scale reparameterization for post-training quantization of vision transformers”, ICCV 2023
>
> [8] Jeon et al., “Genie: show me the data for quantization”, CVPR 2023
>
> [9] Wei et al., “Qdrop: Randomly dropping quantization for extremely low-bit post-training quantization”, ICLR 2022

---

> > ### Comment · Reviewer_L8UP · 2024-11-25
> >
> > Thank you for your detailed rebuttal. I appreciate the time and effort you have put into addressing the concerns raised in the initial review.
> > Overall, the rebuttal has addressed most of my initial concerns. I am now more inclined to support the publication of your manuscript, pending the additional information on scalability.

---

> > > ### Author Response · Authors · 2024-11-25
> > >
> > > Dear Reviewer L8UP,
> > >
> > > Thank you for your thoughtful feedback regarding the scalability of SynQ.
> > > Upon reviewing our rebuttal and the revised manuscript, we realized that there was an error in our reported results.
> > > Specifically, we mistakenly stated the average additional overhead introduced by SynQ as 82.19%, whereas the correct value is 17.81%.
> > > To clarify, the experimental results from Figure 8 are summarized in the table below:
> > >
> > > | Methods | Baseline [sec.] | + SynQ [sec.] | Portion of overhead |
> > > |:---:|:---:|:---:|:---:|
> > > | IntraQ | 13.20 $\pm$ 0.15 | 18.11 $\pm$ 0.22 | **27.10%** |
> > > | HAST | 32.14 $\pm$ 1.29 | 36.87 $\pm$ 2.41 | **12.83%** |
> > > | TexQ | 98.10 $\pm$ 1.97 | 113.40 $\pm$ 2.28 | **13.49%** |
> > >
> > > We sincerely regret this oversight and have updated the rebuttal and manuscript (see Appendix C.2) to reflect the correct values accordingly.
> > > We hope this reply resolves any confusion, and if you have any further questions or require additional clarification, please do not hesitate to let us know.
> > >
> > > Sincerely,
> > >
> > > Authors of Submission 6292

---

### Author Response · Authors · 2024-11-25
**Update of Manuscript**

Dear reviewers,

We sincerely thank all reviewers for their insightful reviews and constructive feedback on our manuscript.

As reviewers highlighted, our paper clearly illustrates and addresses three major challenges in Zero-shot Quantization (reviewers L8UP, bwpy, jAz6, and 7rcv) while emphasizing the relevance of classical low-pass filter techniques in the modern AI era (reviewer jAz6).
We conduct extensive experiments and ablation studies to validate our proposed method, SynQ (reviewers L8UP, bwpy, and jAz6).
Notably, SynQ is the first ZSQ method to comprehensively evaluate performance across both CNNs and ViTs (reviewers bwpy and 7rcv).

In response to the reviewers’ suggestions, we have updated the manuscript and **marked all additions and edits in blue** for your convenience.
Below, we summarize the key changes made in the revised manuscript:

***

### **Additional Experiments and Analyses**
- **[Appendix C.2] Runtime Analysis**
   - We include a runtime analysis in Figure 8, demonstrating that the overhead introduced by SynQ is marginal.
- **[Appendices C.3 and C.4] Broader Analyses of Observations**
   - We conduct additional analyses to show that our observations are not confined to specific conditions but are evident across various scenarios.
Results are in Figures 9, 10, and Table 5.
- **[Appendix C.6] Application of SynQ to Different Noise Optimization Baselines**
   - We extend our experiments to various noise optimization baselines, summarizing the findings in Table 7.
- **[Appendix C.7] Application of SynQ to Zero-shot PTQ**
   - We compare the settings of SynQ and zero-shot PTQ methods, evaluating the effect of applying SynQ in these scenarios. Results are reported in Table 8.
- **[Appendix C.8] Robustness of Low-pass Filter (Idea 1) to Different Types of Noise**
   - We validate the robustness of SynQ’s low-pass filter under various types of noise, as summarized in Table 9.
- **[Appendix C.9] Hyperparameter Analysis on the Size of Synthetic Dataset**
   - We analyze the impact of synthetic dataset size on quantization performance, presenting the results in Figure 11.
- **[Appendix C.10] Visualization of Synthetic Dataset**
   - We include visualizations of the synthetic dataset, which are shown in Figure 12.
- **[Appendix C.11] Hyperparameter Analysis on $\tau$**
   - We conduct additional hyperparameter analyses of the difficulty threshold $\tau$ across different models and datasets in Figure 13.

***

### **Theory and Clarifications**
- **[Sections 3, 4.2, 4.3, 4.5, 5.5, and 5.6] References to Added Appendix Sections**
   - We provide detailed references to the newly added appendix sections in the relevant sections of the manuscript.
- **[Table 1] Header and Caption Revisions**
   - We revise the header and caption of Table 1 to improve clarity and ensure alignment with the manuscript's content.
- **[Section 6] Expansion of Related Work**
   - We expand the discussion of related work to address reviewer feedback and provide further contextual insights.
- **[Appendix C.1] Detailed proof of Theorem 1 (Time Complexity of SynQ)**
   - We include a detailed proof of Theorem 1, focusing on the newly introduced overhead of SynQ.
- **[Appendix D] More details on our experimental setup**
   - We refine the description of our experimental setup, including updates on competitors and baseline methods.

***

We hope these updates, along with our rebuttal, address all the reviewers’ comments thoroughly and enhance the clarity and robustness of our manuscript.
We deeply appreciate your time and consideration once again and look forward to your feedback on the revised version.

With best gratitude,

Authors of Submission 6292

---

### Meta-Review · Area_Chair_GnTZ · 2024-12-18

**Metareview:**

The paper presents SYNQ (Synthesis-aware Fine-tuning for Zero-shot Quantization), a novel framework designed to address the challenges associated with zero-shot quantization (ZSQ) of pre-trained models, particularly in scenarios where training data is inaccessible due to privacy or security concerns. SYNQ tackles three main issues: noise in synthetic datasets, off-target pattern predictions, and misguidance from erroneous hard labels. The proposed method employs a low-pass filter to reduce noise, optimizes class activation map (CAM) alignment to ensure correct image region prediction, and uses soft labels for difficult samples to prevent misguidance. The authors show that SYNQ achieves state-of-the-art accuracy in image classification tasks compared to existing ZSQ methods. This paper is well-written and easy to follow. The observations regarding the three limitations of ZSQ are interesting, and the proposed method appears feasible. The detailed analyses of the three components of SYNQ enhance the persuasiveness of the methodology. The performance is validated through a variety of experiments. Specifically, experiments were conducted to verify the performance of SYNQ by comparing it with various ZDQ baselines on not only CNN-based models but also ViT-based models. While the reviewers had some concerns about the performance in other tasks such as object detection or segmentation, the authors did a particularly good job in their rebuttal. Therefore, all of us have agreed to accept this paper for publication! Please include the additional discussion in the next version.

**Additional Comments On Reviewer Discussion:**

Some reviewers raise the score after the rebuttal.

---

### Decision · Program_Chairs · 2025-01-22

Accept (Poster)